# Coral larvae increase nitrogen assimilation to stabilize algal symbiosis and combat bleaching under increased temperature

Ariana S. Huffmyer[1,2]*, Jill Ashey[2], Emma Strand[2,3], Eric N. Chiles[4,5], Xiaoyang Su[5,6], Hollie M. Putnam[2]

1 School of Aquatic and Fisheries Sciences, University of Washington, Seattle, Washington United States of America, 2 Department of Biological Sciences, University of Rhode Island, Kingston, Rhode Island United States of America, 3 Gloucester Marine Genomics Institute, Gloucester, Massachusetts United States of America, 4 Microbial Biology Graduate Program, Rutgers University, New Brunswick, New Jersey United States of America, 5 Metabolomics Shared Resource, Rutgers Cancer Institute of New Jersey, Rutgers University, New Brunswick, New Jersey United States of America, 6 Department of Medicine, Division of Endocrinology, Robert Wood Johnson Medical School, Rutgers University, New Brunswick, New Jersey United States of America

* ashuffmyer@gmail.com

**Data Availability Statement:** Data and scripts are available on GitHub at https://github.com/AHuffmyer/SymbioticIntegration/releases/tag/v2.1.pub, Zenodo at DOI 10.5281/zenodo.13835295,

## Abstract

Rising sea surface temperatures are increasingly causing breakdown in the nutritional relationship between corals and algal endosymbionts (Symbiodiniaceae), threatening the basis of coral reef ecosystems and highlighting the critical role of coral reproduction in reef maintenance. The effects of thermal stress on metabolic exchange (i.e., transfer of fixed carbon photosynthates from symbiont to host) during sensitive early life stages, however, remains understudied. We exposed symbiotic *Montipora capitata* coral larvae in Hawaiʻi to high temperature (+2.5°C for 3 days), assessed rates of photosynthesis and respiration, and used stable isotope tracing (4 mM $^{13}$C sodium bicarbonate; 4.5 h) to quantify metabolite exchange. While larvae did not show any signs of bleaching and did not experience declines in survival and settlement, metabolic depression was significant under high temperature, indicated by a 19% reduction in respiration rates, but with no change in photosynthesis. Larvae exposed to high temperature showed evidence for maintained translocation of a major photosynthate, glucose, from the symbiont, but there was reduced metabolism of glucose through central carbon metabolism (i.e., glycolysis). The larval host invested in nitrogen cycling by increasing ammonium assimilation, urea metabolism, and sequestration of nitrogen into dipeptides, a mechanism that may support the maintenance of glucose translocation under thermal stress. Host nitrogen assimilation via dipeptide synthesis appears to be used for nitrogen limitation to the Symbiodiniaceae, and we hypothesize that nitrogen limitation contributes to retention of fixed carbon by favoring photosynthate translocation to the host. Collectively, our findings indicate that although these larvae are susceptible to metabolic stress under high temperature, diverting energy to nitrogen assimilation to maintain symbiont population density, photosynthesis, and carbon translocation may allow larvae to avoid bleaching and highlights potential life stage specific metabolic responses to stress.

and Open Science Framework (OSF) at DOI 10.
17605/OSF.IO/TE6S2. Metabolomics data files (.
raw and .mzXML files) and data underlying figures
are also available on OSF at DOI 10.17605/OSF.IO/
TE6S2.

**Funding:** This research was supported by the
National Science Foundation Ocean Sciences
Postdoctoral Fellowship (2205966 to ASH),
National Science Foundation Rules of Life-
Epigenetics (EF-1921465 to HMP), and a gift of the
Washington Research Foundation to the University
of Washington eScience Institute (eScience Data
Science Postdoctoral Fellowship award to ASH).
The funders had no role in study design, data
collection and analysis, decision to publish, or
preparation of the manuscript.

**Competing interests:** The authors have declared
that no competing interests exist.

**Abbreviations:** AGC, automatic gain control; BCA,
bicinchoninic acid; dpf, days post fertilization; FSW,
filtered seawater; GDH, glutamate dehydrogenase;
GS, glutamine synthetase; HESI, heated
electrospray ionization; hpf, hours post fertilization;
OSF, Open Science Framework; PCA, principal
components analysis; PLS-DA, partial least
squares discriminant analysis; PPP, pentose
phosphate pathway; TCA, tricarboxylic acid; VIP,
variable importance in projection.

## Introduction

Reef-building corals are responsible for the foundational structure of coral reef ecosystems, made possible by the nutritional symbiosis between intracellular dinoflagellate algae (family Symbiodiniaceae [1]) and their scleractinian coral hosts [2,3]. Symbionts are housed within the coral gastrodermal tissues in cellular compartments (i.e., symbiosomes) and conduct photosynthesis to produce fixed carbon photosynthates (e.g., sugars, lipids, amino acids) that are translocated to the coral host to meet metabolic needs [4–6]. The coral host provides nitrogen (e.g., ammonium $NH_4+$ and amino acids) and inorganic carbon (e.g., $CO_2$) metabolic products to the symbiont that are required for photosynthesis and population growth [7–9].

The delicate symbiotic nutritional balance between Symbiodiniaceae and corals can be challenged by rapidly changing physical environments like those associated with marine heatwaves, which are increasing in frequency and severity, ultimately leading to global-scale coral bleaching events [10–12]. Coral bleaching under high temperature is characterized by the loss of the Symbiodiniaceae, the coral's primary source of nutrition via their translocated photosynthates [13–15], which can result in mass mortality [11,16–18]. The mechanisms underlying the onset and progression of coral bleaching are complex and a topic of intense scientific study (see a recent review in [19]). Research in the last 2 decades has demonstrated that oxidative stress [19,20] and carbon and nitrogen imbalance models may explain dysbiosis that occurs under bleaching conditions [19,21–25].

Carbon-nitrogen exchange is a core process regulating symbiotic stability [21,22,26]. Host control of nitrogen is critical to maintaining carbon translocation [9,22,26,27]. For example, in a stable symbiotic state, nitrogen limitation by the coral host allows for excess photosynthates not required by the symbiont to be translocated to the host [4,7,22,26,28]. When released from nitrogen limitation, symbiont population growth increases, requiring greater photosynthate retention within the symbiont, thus less is translocated to the host [7,9,27,29]. Under high temperatures, increased metabolic demand in the host and increased metabolism of lipid reserves and amino acids produces elevated nitrogenous waste, increasing nitrogen availability [9,21,28,30]. As symbiont populations grow and retain more fixed carbon and subsequently translocate less to the host [28,29], there is depletion of host energy resources [31,32], and therefore less energy available to actively sequester nitrogen [21], exacerbating nitrogen imbalance and destabilizing symbiosis [22,28]. Conceptual models propose that the decoupling of carbon and nitrogen cycling can lead to breakdown of the symbiotic relationship under thermal stress [28]. Recent experimental evidence from adult *Exaiptasia* anemones and reef building corals demonstrate that the supply of carbon regulates nitrogen availability to the symbiont, supporting the role of coupled carbon-nitrogen metabolism in generating a stable symbiotic state under ambient conditions [21,22]. There is a critical need for experimental research to test conceptual models and investigate the role of nutrient cycling on the breakdown of the symbiotic relationship under elevated temperatures. Further, key knowledge gaps remain on how these processes vary across life stages, especially in early life stages during which the symbiotic relationship is established. Therefore, in this study we used stable isotope metabolomic tracing to investigate carbon and nitrogen metabolic feedback and shifts in symbiotic nutritional exchange under thermal stress in symbiotic coral larvae.

Evaluating the mechanisms of maintaining functional nutritional symbiosis is particularly crucial during larval stages of corals. In the face of increasingly frequent coral bleaching events, successful settlement and recruitment of reef-building coral larvae determines the survival of the reef ecosystem [11,33–35]. Development, dispersal, and settlement are energetically costly processes [36–38] and larvae heavily depend on stored lipid reserves to meet this demand [39–42]. However, offspring in vertically transmitting species have symbiont-derived nutrition

(e.g., translocated carbon and fatty acids) that can support this energy demand [39,40,42,43]. Because larvae experience energetic challenges during development [36,44,45], it is important to understand whether they are more susceptible to thermal stress-induced dysbiosis in this crucial period [37,46]. Further, because coral reproduction often occurs outside the peak annual temperature (e.g., reproduction in June-July in *Montipora capitata* with peak temperatures in August-September in Hawai'i [47–49]), it is important to understand how symbiotic relationships are affected by increasing, yet sublethal or sub-bleaching, temperatures that occur during reproductive seasons.

For a dominant vertically transmitting spawning species in Hawai'i, *Montipora capitata*, symbiotic larvae are competent to settle within 7 days post fertilization [50,51], representing a critical window in which thermal stress could impact settlement. Therefore, in this study we tested the hypothesis that a sublethal thermal stress in the pre-competency window (+2.5°C for 3 days from 4 to 7 days post fertilization) impacts larval symbiotic nutritional exchange and carbon metabolism. To track nutrient translocation and characterize flux through key metabolic pathways, we conducted a $^{13}C$ metabolomic tracing (4 mM $^{13}C$ labeled sodium bicarbonate) experiment (Fig 1). With this method, $^{13}C$ is taken up by the symbiont in the form of dissolved inorganic carbon, fixed through photosynthesis, and translocated to the host. Therefore, the presence of labeled photosynthate metabolites provides information on the content of translocated carbon and the utilization of central metabolism pathways by the host (Fig 2). We hypothesized that exposure to increased temperatures would reduce the translocation of photosynthates and result in larval bleaching.

## Materials and methods

### Ethics statement

Coral collections in this study were approved under Special Activity Permit no. 2021–41 issued by the Hawai'i Department of Land and Natural Resources (Division of Aquatic Resources, Honolulu, HI).

### Gamete fertilization and embryo rearing

To investigate how short-term increases in temperature impact metabolism in early life history, we exposed larvae of the broadcast spawning species *Montipora capitata* to high temperature treatments for 3 days (approximately 3.5 to 7 days post fertilization; Fig 1E) at the Hawai'i Institute of Marine Biology (HIMB) in Kāne'ohe Bay, O'ahu, Hawai'i. *Montipora capitata* is a dominant reef-building species (Fig 1A) in this location and displays vertical symbiont transmission that provides Symbiodiniaceae to offspring through the egg. Gametes (egg-sperm bundles) were collected from the natural population in Kāne'ohe Bay (mid-bay near reef #11; 21° 26' 56" N, 157° 47' 45" W) between 21:00 and 22:00 on 11 June 2021. Buoyant bundles were scooped from the surface using mesh bottom containers, pooled in 5 gal buckets, and gently mixed as described in [47]. Briefly, 5 ml aliquots of bundles were added to 50 ml Falcon tubes filled with 1 μm filtered seawater (FSW) and gently rocked every 20 to 30 min. After 1.5 h, fertilized eggs were then separated from sperm, rinsed gently in 1 μm FSW, and added to 20 L polycarbonate tanks (*n* = 12 tanks) filled with 1 μm FSW. Cultures of developing embryos in tanks were checked every 1 to 2 h and dead material or debris was removed.

Tanks were equipped with 7.6 cm diameter banjo filters (Fig 1B; 153 μm mesh) on their outflows and supplied with flow-through sand-filtered seawater (approx. 90 ml min$^{-1}$). Larval culture tanks were partially submerged in larger water baths to control temperature (Fig 1B; *n* = 6 tanks per water bath, *n* = 2 water baths). Tanks were covered with shade cloth such that light was approx. 100 μmol photons m$^{-2}$ s$^{-1}$ (PAR sensor, MQ-510 Apogee Instruments, Logan, Utah,

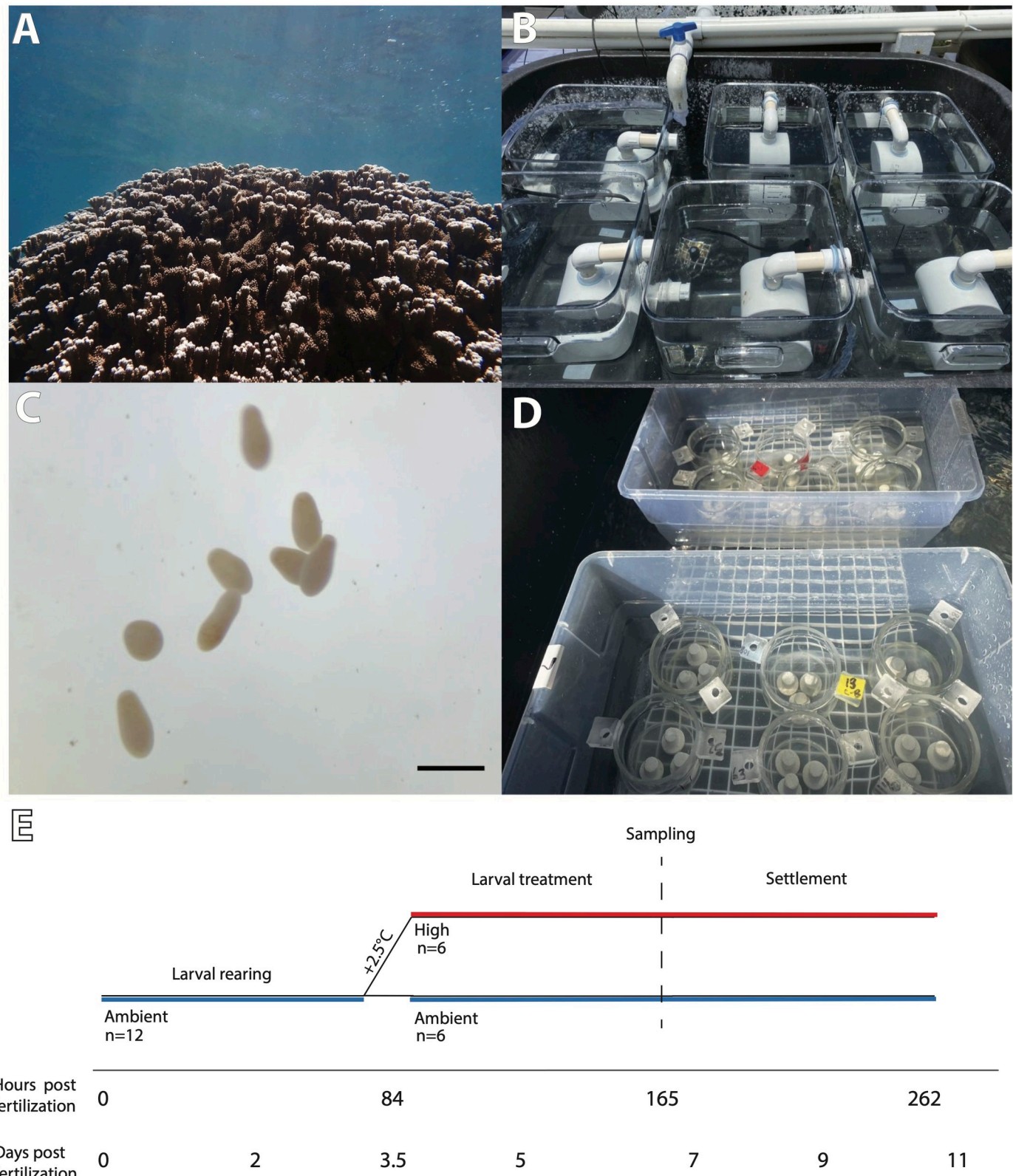

**Fig 1. (A) Adult *Montipora capitata* colony; (B) larval rearing tanks equipped with banjo filters submerged in water bath; (C) *Montipora capitata* planulae, scale bar indicates 1 mm; (D) larval settlement chambers with aragonite settlement plugs; (E) timeline of experiment.** Larvae were reared at ambient from 0–3.5 days post fertilization (dpf) (0–84 h post fertilization; hpf) with high (+2.5°C) and ambient treatments beginning at 3.5 dpf (84 hpf). Replication indicates the number of larval tanks at each treatment. Sampling of larvae occurred at approximately 7 dpf (165 hpf). Settlement was measured from approximately 7–11 dpf (165–262 hpf).

USA). Temperature was recorded within the larval cultures every 15 min using Hobo temperature loggers (Onset U22, Onset Computer Corp. Bourne, Massachusetts, USA; $n = 3$ loggers per temperature treatment in 3 randomly selected tanks per treatment). At approximately 24 h post fertilization (hpf), flow rate was increased to approx. 180 ml min$^{-1}$. Mean temperature (± std. dev.) from fertilization through 84 hpf was 26.5 ± 0.5°C ($n = 2,916$) (S1 Fig). On 14 June 2021 (60 hpf), planula larvae from all tanks were pooled and tanks were cleaned by rinsing with freshwater and drying in preparation for the initiation of temperature treatments. The larvae were then allocated to each of the 12 larval tanks at a density of 1 larva per ml (20,000 larvae in each replicate 20 L culture tank). Active swimming behavior was observed at approx. 72 hpf (Fig 1C).

## Larval thermal exposure

Ambient sand-filtered seawater was delivered into two 113 L insulated Igloo coolers as header tanks. Temperature was manipulated in one header tank with submersible titanium heaters (300W, Finnex, Chicago, Illinois, USA) and an Apex Neptune AquaController system (Neptune Systems, Morgan Hill, California, USA) to generate a high temperature treatment that supplied water to larval tanks ($N = 6$; "high" temperature treatment). The second tank held ambient temperature seawater that supplied larval tanks ($N = 6$; "ambient" temperature treatment). The high temperature treatment was further controlled using 2 additional heaters in the water bath surrounding the high treatment larval tanks with recirculating pumps to mix the water bath. Water (ambient or heated) was pumped from the header tanks to each larval tank using magnetic drive pumps (2400 GPH, Pondmaster, Kissimmee, Florida, USA) that delivered water through tubing into each tank. Temperature treatment was started on 15 June 2021 at 08:00 h and reached treatment temperature by 11:00 h (approx. 84 hpf; Fig 1E). The high temperature treatment raised the temperature by approx. 2.5°C above ambient, maintaining the same diel fluctuations (approx. 1.3°C diel range). Maximum diel temperatures reached 30.9°C in the high treatment and 27.8°C in the ambient treatment (S1 Fig). During the larval thermal exposure, mean temperature (± std. dev.) in the ambient treatment was 26.7 ± 0.4°C ($n = 1,482$) and 29.2 ± 0.6°C ($n = 1,482$; S1 Fig) in the high temperature treatment. Mean (± std. dev.) light was 60 ± 28 μmol photons m$^{-2}$ s$^{-1}$ ($n = 48$), mean pH$_{Total}$ was 7.92 ± 0.02 ($n = 48$), and mean salinity was 34.7 ± 0.31 psu ($n = 48$) in larval tanks as measured across one representative day during the treatment period.

Larval survivorship was measured every 12 h (between 08:00 and 09:30 h and 22:00 to 23:30 h each day) by gently mixing larvae in each tank, aliquoting six 25 ml samples per tank, and counting the number of larvae per ml on a dissecting microscope. Larval survivorship was calculated as mean larvae per ml ($n = 6$ counts per tank). The effect of time and treatment on larval survival was tested with a two-way analysis of variance (ANOVA) in R Statistical Programming v4.2.2 [52]. Residual normality and homogeneity of variance were assessed using quantile-quantile plots and Levene's tests in the *car* package, respectively [53]. All data and scripts are openly accessible at Zenodo at 10.5281/zenodo.13835295 and Open Science Framework (OSF) at 10.17605/OSF.IO/TE6S2.

## Larval response to treatment

**Larval metabolic rates.** Sampling occurred on 18 June 2021 from 16:00 to 20:00 h (approx. 7 days post fertilization; Fig 1E). Live larvae were collected from each larval tank on

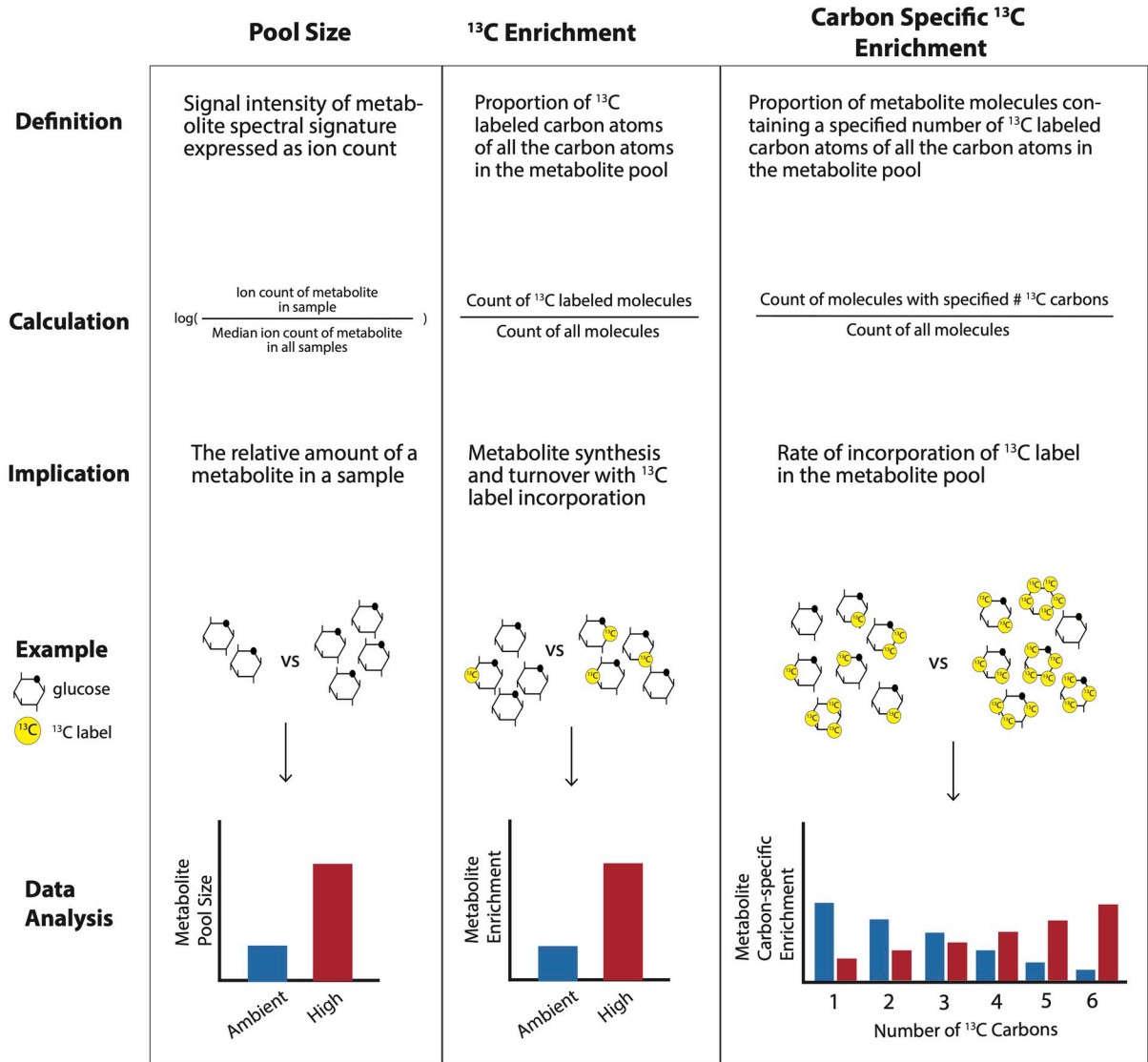

**Fig 2. Overview of 3 data types analyzed in this study: pool size (left column), $^{13}$C enrichment (middle column), and carbon specific $^{13}$C enrichment (right column).** Pool size is the signal intensity of a metabolite spectral signature expressed as ion count and calculated as log-transformed median-normalized ion counts for analysis. Pool size indicates the relative amount of metabolite in a sample. In this study, we compared pool size for metabolites between temperature treatments. $^{13}$C enrichment is the proportion of $^{13}$C labeled carbon atoms of all the carbon atoms in the metabolite pool and $^{13}$C incorporation provides information on metabolite synthesis and turnover. In this study, we compared enrichment for metabolites between temperature treatments. Finally, carbon specific $^{13}$C enrichment is the proportion of metabolite molecules containing a specified number of $^{13}$C labeled carbon atoms of all carbon atoms in the metabolite pool and provides information on the rate of $^{13}$C incorporation into the metabolite pool. For example, in a glucose molecule with 6 carbon atoms, we analyzed whether the number of carbons with a $^{13}$C label in glucose metabolites shifts between treatments.

18 June 2021 at 16:00 h (77 h exposure) for measurement of metabolic rates. Respiration and photosynthetic rates were measured in larvae from both high and ambient temperature treatments at both 30°C and 27°C in a reciprocal exposure. Temperature was controlled using benchtop incubators (MyTemp Digital Heat & Cool Incubator; BenchMark Scientific, Sayreville, New Jersey, USA) equipped with LED aquarium lighting (Hydra 16HD, AquaIllumination, Bethlehem, Pennsylvania, USA) set to approx. 500 μmol photons $m^{-2}$ $s^{-1}$. Respiration (oxygen consumption) and photosynthesis (oxygen production) were measured using 2 Oxy-10 Sensor Dish Readers (4-channel SensorDish Reader; PreSens Precision Sensing, Regensburg, Germany; cat. 200001058) in combination with two 80 μl 24-well glass microplates (product CH25000 Loligo Systems, Viborg, Denmark) with each containing a PreSens oxygen sensor spot. Microplates were filled with 1 μm FSW and sensors were allowed to condition for 15 min at room temperature prior to oxygen measurements. Oxygen sensor spots were calibrated with a single point calibration using 100% air saturated seawater according to manufacturer's instructions. Larvae were rinsed in 1 μm FSW held at the treatment temperature and pools of 6 larvae were added to 80 μl microplate wells filled with oxygen saturated 1 μm FSW at the treatment temperature, with $n = 2$ wells per replicate larval exposure tank per microplate ($n = 20–22$ larval pools per treatment group). Two wells of each plate were filled with oxygen saturated 1 μm FSW at the treatment temperature to serve as blanks. Wells were then sealed using glass coverslips and moved into incubators at the specified temperature.

Net photosynthesis (i.e., oxygen production in μmol $L^{-1}$), was measured first under light conditions for 30 to 40 min at approx. 500 μmol photons $m^{-2}$ $s^{-1}$ with oxygen measurements collected every 15 s. The lights were then turned off and oxygen consumption (i.e., light-enhanced dark respiration, referred to hereafter as "respiration"; [54]) was measured for an additional 20 to 30 min in the dark. Metabolic rates were extracted using localized linear regressions in the *LoLinR* package [55] using the percentile rank method and alpha set to 0.4 in R Statistical Programming v4.2.2 [52]. Respiration and net photosynthesis rates were corrected by normalizing to the well volume and then subtracting blank values and dividing by the number of individuals in each well for units of nmol $O_2$ individual$^{-1}$ min$^{-1}$. Metabolic rates were then normalized to mean larval size from the respective treatment group (nmol $O_2$ mm$^{-3}$ min$^{-1}$), which was measured as described in the larval size section below. We calculated gross photosynthesis by adding net photosynthesis to respiration rates (oxygen produced + oxygen consumed) and calculated photosynthesis: respiration ratio (P:R) as gross photosynthesis divided by respiration for size-normalized metabolic rates. Size normalized respiration, net photosynthesis, gross photosynthesis, and P:R were analyzed using two-way ANOVA tests with the main effects of exposure history and assay temperature. Residual normality and homogeneity of variance were assessed using quantile-quantile plots and Levene's tests in the *car* package, respectively [53].

**Physiological sampling.** Samples were collected on 18 June 2021 at 18:00 h (79 h exposure) for physiological assays by filling 2 ml cryotubes with 1.5 ml pools of larvae ($n = 100–300$ larvae per pool), centrifuged at 13,000 g for 1.5 min to remove seawater, snap frozen in liquid nitrogen, and stored at −80°C ($n = 2$ sample tubes per larval tank; $N = 24$ total samples). Pools of larvae (20 to 30 larvae per tank) were fixed in 4% paraformaldehyde stored at 4°C for analysis of larval size.

**Stable isotope labeling.** We utilized carbon-13 stable isotope ($^{13}$C) tracing in conjunction with mass spectrometry to track exchange between the coral host and algal symbiont and metabolic flux in response to high temperature treatments following methods described by [56,57]. On 18 June 2021 at 20:00 h (81 h exposure), larvae were incubated for 4.5 h in either labeled (4 mM NaH$^{13}$CO$_3$ sodium bicarbonate; Cambridge Isotope Laboratories, Tewksbury, Massachusetts, USA), unlabeled (4 mM NaHCO$_3$ sodium bicarbonate), or dark labeled (no

light with 4 mM NaH$^{13}$CO$_3$ sodium bicarbonate) conditions in 20 ml scintillation vials in the light (approx. 500 μmol m$^{-2}$ s$^{-1}$) controlled in benchtop incubators at treatment temperature. Solutions were made by adding either labeled or unlabeled sodium bicarbonate to 1 μm FSW to obtain a 4 mM concentration at a pH of 7.9 to 8.0 [56,57]. Larvae from the ambient rearing treatment were incubated at 28.3 ± 0.3˚C (mean ± std. dev.; $n = 5$) and larvae from the high rearing treatment were incubated at 31.0 ± 0.5˚C ($n = 5$) to quantify metabolic responses to respective peak temperatures reached in each treatment. Pools of larvae (approx. 200 larvae per pool) were added to either labeled seawater ($n = 6$ vials per temperature) or unlabeled seawater ($n = 6$ vials per temperature). Two vials contained a sample of larvae pooled from all larval tanks within the respective temperature treatment and wrapped in aluminum foil in labeled solution to serve as a dark control ($n = 2$ vials per temperature). The purpose of the dark control was to verify that the carbon in photosynthates and photosynthate metabolism intermediate metabolites were sourced from symbiont uptake of inorganic labeled carbon and fixation through photosynthesis and subsequent translocation to the coral host [56]. Caps on vials were loosely secured to allow for air exchange. After addition of the solutions in larval vials, vials were placed in incubators at the respective temperatures and incubated for 4.5 h. Each hour, vials were mixed by gently rocking for approx. 5 s. Larvae were sampled at the end of the incubation by pouring larvae through sterile 100 μm cell strainers to remove seawater, rinsed 3 times in 1 μm FSW, and snap frozen in liquid nitrogen. Samples were stored at −80˚C until processing.

## Settlement

To assess larval settlement, larvae were added into settlement chambers at the treatment temperatures at 14:00 h on 19 June 2021. Transparent acrylic settlement chambers (200 ml) were constructed to be open on top, with the bottom covered with 153 μm mesh to allow for water exchange into the chambers. Aragonite plugs ($n = 3$ plugs per chamber, 2.2 cm; Ocean Wonders, Decorah, Iowa, USA; soaked in seawater for 1 h before use) were placed halfway submerged in temperature manipulated seawater in a large water bath tank on plastic crate racks (Fig 1D). Larval pools ($n = 100$ larvae per pool) were added into each of 2 chambers per larval tank ($n = 12$ chambers per temperature treatment). Settled larvae (i.e., primary polyps) were identified by the presence of an oral disk, septa formation, and/or polyp structures with attachment to the substrate. Settlement was measured every 1 to 2 days from 19 June 2021 through 26 June 2021 between 11:00 h and 14:00 h daily by counting the number of primary polyps. Temperature and environmental conditions were controlled as described above. During the settlement period, mean temperature (± std. dev.) in the ambient treatment was 27.2 ± 0.6˚C ($n = 3,051$) and was 29.2 ± 0.5˚C ($n = 3,052$; S1 Fig) in the high temperature treatment. Settlement was calculated as a proportion of the number of primary polyps divided by the total number of starting larvae. The effect of time and treatment on larval settlement was tested with a two-way ANOVA in R Statistical Programming v4.2.2 [52] with residual normality and homogeneity of variance assessed using quantile-quantile plots and Levene's tests in the *car* package, respectively [53].

## Physiological quantification and sample processing

Samples were processed for physiological parameters including symbiont cell density, chlorophyll content, and total carbohydrates, which were normalized to total soluble host protein. Samples ($n = 12$ per treatment, $n = 2$ samples per larval tank) were thawed on ice and 600 μl of 1× PBS buffer were added to dilute the tissue concentration. Samples were then homogenized with an immersion tip homogenizer (Benchmark D1000) for 30 to 45 s. Homogenized tissue

(500 μl) was then transferred to a new tube and designated as the holobiont fraction. The remaining homogenate (100 μl) was transferred to a new tube and centrifuged for 90 s at 2,000 g to pellet the symbiont fraction and split into 2 aliquots for measuring symbiont cell density and chlorophyll concentration. The supernatant containing the host fraction was then transferred to a new tube and the volume was recorded. The homogenizer was cleaned with 10% bleach, 70% ethanol, and deionized water in between samples.

**Soluble protein.** Host fractions were thawed on ice and vortexed for 15 s. Soluble protein was quantified using the Pierce BCA Protein Assay Kit (Thermo Scientific, Bothell, Washington, USA) with bovine serum albumin as the standard, and 25 μl of sample were added to a 96-well plate in triplicates. Approximately 200 μl of working reagent (bicinchoninic acid (BCA) reagent mix of 50:1 of components A and B) were added to each well and mixed with the sample for colorimetric detection of absorbance at 562 nm. The 96-well plate was incubated at 37°C for 30 min and cooled to room temperature prior to reading on a plate reader (Synergy HTX Multi-Mode Microplate Reader, model S1LFA, Agilent Bio Tek, Santa Clara, California, USA).

**Larval size.** Fixed larvae were imaged to quantify larval size. Subsets of 20 to 30 larvae per sample ($n$ = 6 larval pools per treatment; $n$ = 120 larvae per treatment) were placed in a microscope dish and images were taken with OMAX Digital Microscope Camera (Model A35180U3, Kitchener ON Canada) on a OMAX dissecting light microscope with an OMAX 0.01 mm stage micrometer (Model A36CALM1). Larval length and width of the longitudinal and transverse axes, respectively, were measured for each larva using ImageJ software [58]. Larval volume was calculated by applying the equation for volume of an elliptical sphere, $V = \frac{4}{3}\pi ab^2$, where $a$ is $\frac{1}{2}$ width and $b$ is $\frac{1}{2}$ length [59]. To calculate measurement uncertainty, we conducted 20 repeated measurements of a known scale bar length (0.100 mm) and calculated the mean differences between measurements and known length. We also calculated the known volume using a known length (0.01 mm) and width (0.01 mm) using the equation for an elliptical sphere described above. Measurement uncertainty was calculated as 0.0007 ± 0.0005 mm (mean ± standard deviation) in length and 0.00001 ± 0.000009 mm$^3$ (mean ± standard deviation) in volume.

**Symbiont cell density.** The symbiont fraction was thawed on ice and vortexed for 15 s. Symbiodiniaceae cell density of each sample was quantified with 6 technical replicate counts on a hemocytometer (Hausser Scientific, Horsham, Pennsylvania, USA). Symbiont cell density was normalized to host protein content (cells μg host protein$^{-1}$).

**Chlorophyll content.** To quantify chlorophyll content, 200 μl from the symbiont fraction aliquot were centrifuged at 13,000 g for 3 min. The supernatant was removed and 1 ml of 100% acetone was added to the pellet. The sample was then vortexed for 15 s and stored at 4°C for 24 h in the dark. After the incubation period, the sample was vortexed again for 15 s and centrifuged at 13,000 g for 3 min. Two 200 μl replicates per sample were read on a 96-well microplate reader (Synergy HTX Multi-Mode Microplate Reader, model S1LFA, Agilent Bio Tek, Santa Clara, California, USA) at 630 and 663 nm. Chlorophyll-$a$ and $c2$ were calculated according to [60] as $11.43E_{663}-0.64E_{630}$ and $27.09E_{630}-3.63E_{663}$, respectively. Resulting values were divided by 0.584 to correct for path length of the plate and standardized to host protein content (μg chlorophyll μg host protein$^{-1}$) and Symbiodiniaceae cell density (μg chlorophyll cell$^{-1}$). Total chlorophyll content was calculated as the sum of chlorophyll-$a$ and $c2$ concentrations.

**Total carbohydrate content.** Host fractions for each sample were analyzed for total carbohydrates following the phenol-sulfuric acid method modified for use in 96-well plates [61–63], and 25 μl of each sample (host fractions of $n$ = 24 samples) were added to 975 μl of

deionized water in 5 ml tubes. Carbohydrates were extracted by adding 44 µl of phenol and 2.5 ml of sulfuric acid to each sample or L-(-)-Glucose standards (Sigma-Aldrich cat. G5500) and allowed to incubate for 30 min. Following incubation, 200 µl of each sample and each standard were then added to 96-well plates in triplicate. Absorbance of each well was then read on a plate reader (Synergy HTX Multi-Mode Microplate Reader, model S1LFA, Agilent Bio Tek, Santa Clara, California, USA) at 485 nm. Carbohydrate concentration (mg mL$^{-1}$) was determined against a standard curve of the L-(-)-Glucose standards as a function of absorbance at 485 nm and normalized to total host soluble protein (µg carbohydrate µg host protein$^{-1}$) and symbiont cell density (µg carbohydrate cell$^{-1}$).

**Physiology data analysis.** For all physiological analyses, the effect of treatment on each metric was tested with two-sample Welch $t$ tests in R Statistical Programming v4.2.2 [52]. Residual normality and homogeneity of variance were assessed using quantile-quantile plots and Levene's tests in the *car* package, respectively [53]. Holobiont carbohydrate content per Symbiodiniaceae cell (µg cell$^{-1}$) was the only metric analyzed using a nonparametric Wilcoxon rank sum exact test, due to violation of homogeneity of variance.

## Metabolomics

**Metabolite extraction.** Snap frozen larvae stored at −80˚C were homogenized in 1.5 ml tubes using an immersion homogenizer (Benchmark D1000) and separated into host and symbiont fractions using a bench-top centrifuge at 4˚C at 9,000 g for 3 min. Host supernatant fractions were added to ice cold dounces with 500 µl of metabolomic extraction buffer (40:40:20 ratio of methanol, acetonitrile, and ultrapure water with 0.5% [v/v] formic acid). Samples were allowed to incubate in the extraction buffer on dry ice for 5 min. Following extraction, samples were homogenized for 1 min on dry ice with the glass dounce. Samples were then transferred to new 1.5 ml tubes and centrifuged at 4˚C at 15,000 g for 10 min. After centrifugation, 500 µl of supernatant were moved to new 1.5 ml tubes, 44 µl of 15% ammonium bicarbonate were added, the tubes were vortexed for 15 s, and 100 µl of the extracted samples were then aliquoted into duplicate 2 ml autosampler vials. All materials and equipment were cleaned with 10% bleach, 70% ethanol, and deionized water and autoclaved in between batches of extractions ($n$ = 6 samples extracted per batch). Metabolites were extracted from $^{12}$C control samples followed by $^{13}$C dark incubation and $^{13}$C light incubation samples to prevent $^{13}$C contamination across isotope treatments. Samples were extracted in a random order within isotope treatment.

**Metabolite quantification and spectrometry analysis.** The Hydrophilic Interaction Liquid Chromatography (HILIC) separation was performed on a Vanquish Horizon UHPLC system (Thermo Fisher Scientific, Waltham, Massachusetts, USA) with XBridge BEH Amide column (150 mm × 2.1 mm, 2.5 µm particle size, Waters, Milford, Massachusetts, USA) using a gradient of solvent A (95%:5% H$_2$O:acetonitrile with 20 mM acetic acid, 40 mM ammonium hydroxide, pH 9.4), and solvent B (20%:80% H$_2$O:acetonitrile with 20 mM acetic acid, 40 mM ammonium hydroxide, pH 9.4). The gradient was 0 min, 100% B; 3 min, 100% B; 3.2 min, 90% B; 6.2 min, 90% B; 6.5 min, 80% B; 10.5 min, 80% B; 10.7 min, 70% B; 13.5 min, 70% B; 13.7 min, 45% B; 16 min, 45% B; 16.5 min, 100% B and 22 min, 100% B [64]. The flow rate was 300 µl per min. Injection volume was 5 µl and column temperature was 25˚C with the autosampler temperature set to 4˚C.

Full scan mass spectrometry analysis was performed on a Thermo Q Exactive PLUS mass spectrometer with a heated electrospray ionization (HESI) source which was set to a spray voltage of −2.7 kV under negative mode and 3.5 kV under positive mode. The sheath, auxiliary, and sweep gas flow rates of 40, 10, and 2 (arbitrary unit), respectively. The capillary

temperature was set to 300˚C and the aux gas heater was 360˚C. The S-lens RF level was 45. The m/z scan range was set to 72 to 1,000 m/z under both positive and negative ionization mode. The automatic gain control (AGC) target was set to 3e$^6$ and the maximum IT was 200 ms. The resolution was set to 70,000.

The metabolite abundance data were obtained from the first spectrometer (MS$^1$) full scans using the MAVEN software package [65]. The compound identification was verified by accurate mass and retention time matching to an in-house library derived from pure chemical standards. Base peaks and labeled fractions were verified with manual peak picking using a mass accuracy window of 5 ppm. The isotope natural abundance was corrected using AccuCor [66]. Prior to running the samples, the Thermo Q Exactive PLUS mass spectrometer was evaluated for performance readiness by running a commercially available standard mixture (Thermo Scientific Pierce LTQ ESI Positive Ion Calibration Solution Cat # 88322) and an in-house standard mixture (lactate, leucine, isoleucine, malate, glucose-6-phosphate, fructose-6-phosphate, NAD, ATP) to assess the mass accuracy, signal intensities and the retention time consistency. All known metabolites in both the commercial and in-house calibration mixtures are detected within 5 ppm mass accuracy. Reagent blank samples matching the composition of the extraction solvent without addition of samples are used in every sample batch to assess background signals and ensure there was no carryover from one run to the next. In addition, the sample queue was randomized with respect to sample treatment to eliminate the potential for batch effects.

**Metabolomics data description.** Three metabolomic response variables were analyzed in this study: pool size, metabolite $^{13}$C enrichment, and carbon-specific $^{13}$C enrichment (Fig 2). First, pool size indicates the relative amount of a particular metabolite in a sample. Pool size is quantified by peak intensity values for each metabolite that were median-normalized (i.e., metabolite intensity values divided by the median peak intensity of all metabolites within that sample) to account for variation in total biomass between samples, followed by transformation (log + 1) prior to analyses. Second, metabolite $^{13}$C enrichment is calculated as the proportion of $^{13}$C atoms of total carbon atoms of the respective metabolite. Third, carbon-specific $^{13}$C enrichment is calculated as the proportion of molecules of a metabolite with a specified number of $^{13}$C atoms of the entire pool of the metabolite. For example, this calculation would indicate the proportion of glucose molecules that have zero, one, two, three, four, five, or six $^{13}$C labeled carbon atoms (out of 6 total carbon atoms in a glucose molecule).

Differences in pool size in these temperature treatments could arise from 2 possible scenarios (Fig 3A x-axis). Accumulation of a particular metabolite could be due to a higher production or synthesis of a metabolite while downstream usage remains constant, and/or production or synthesis of a metabolite remains constant while downstream usage is reduced. To determine which process contributes to differences in metabolite pool size, we examined isotopic enrichment to provide information on the rate of metabolite production and turnover (i.e., the relative balance between metabolite production and downstream usage). Shifts in metabolite enrichment between treatments in this study could arise from increased contribution of photosynthetically fixed carbon to the synthesis of the metabolite relative to synthesis from host-derived carbon, and/or increased metabolite turnover via production of a metabolite along with increased downstream usage (Fig 3A y-axis). To determine whether shifts in metabolite turnover were occurring, we considered both metabolite pool size and metabolite enrichment (Fig 3). For example, a metabolite that has increased pool size, but no concurrent increase in enrichment, is likely resulting from accumulation due to reduced downstream metabolism (Fig 3Aiv). In contrast, a metabolite with stable pool size and increased enrichment is likely to be under high turnover due to increased production and continued downstream usage (Fig 3Ai). A metabolite with both increased pool size and elevated enrichment

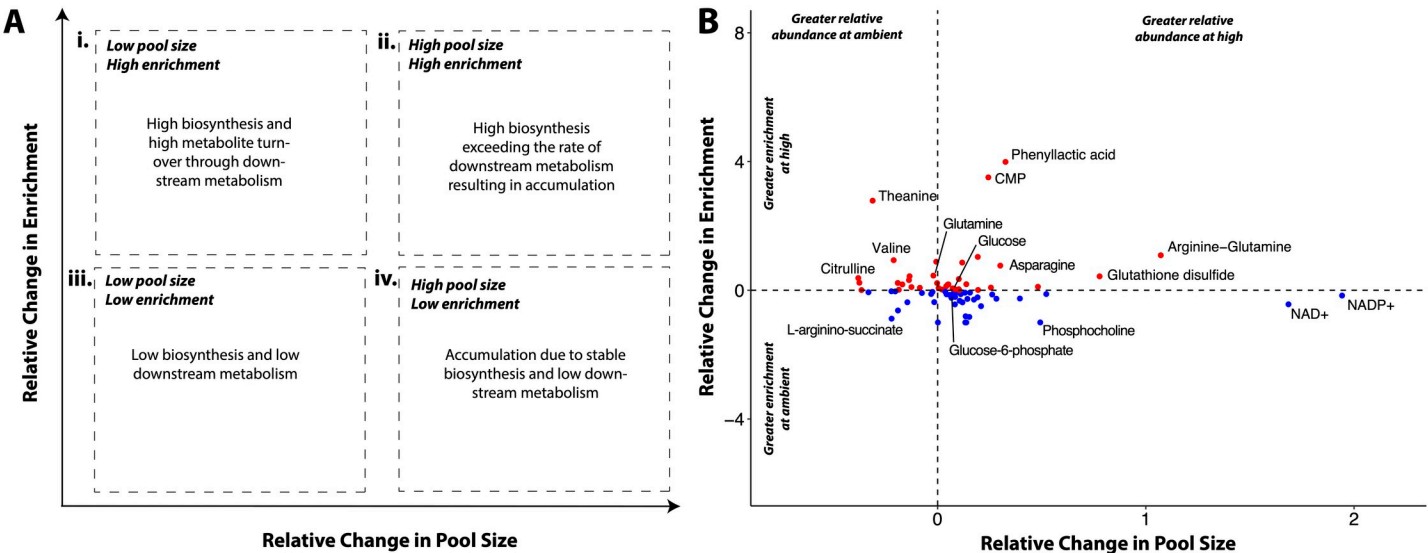

**Fig 3. (A) Framework with general interpretations of four scenarios regarding metabolite pool size and $^{13}$C enrichment.** If a high pool size occurs with low enrichment, that metabolite is likely accumulating because of low biosynthesis and downstream metabolism (iv). In contrast, if a high pool size occurs with high enrichment (ii), that metabolite may be generated at a high rate exceeding that of downstream metabolism. Metabolites displaying both low pool size and enrichment may be undergoing both low biosynthesis and low downstream metabolism (iii). Finally, metabolites with low or stable pool size but high enrichment indicate a high rate of metabolite turnover due to biosynthesis that meets the rate of downstream metabolic activity (i). In this study, we examined relative differences in metabolite pool size and enrichment between temperature treatments. (B) Mean relative change in pool size (x-axis) and enrichment (y-axis) for detected metabolites. Points >0 in relative change in pool size (x) are metabolites that are in greater concentrations at high temperatures; Points >0 in relative change in enrichment (y) are metabolites more enriched at high temperature. Dotted lines separate the 4 quadrants of the framework in (A). Labels added to representative metabolites in each quadrant. The data underlying this figure can be found at 10.5281/zenodo.13835295.

could be undergoing high rates of synthesis that overwhelm the capacity for downstream usage, although turnover remains high (Fig 3Aii). In our study system, increased enrichment of photosynthates indicates relatively greater photosynthetically fixed $^{13}$C incorporation. For downstream metabolites, increased enrichment indicates that photosynthetically fixed carbon is utilized in the respective pathway and that the downstream metabolite is undergoing elevated turnover. Given this construct for interpreting metabolic responses, we examined metabolites involved in key metabolic pathways to infer how metabolism was altered by temperature exposure (Fig 3B).

In this study, we analyzed metabolomic response for key pathways of interest involved in central carbon and nitrogen metabolism to test the hypothesis that exposure to thermal stress decreases carbon translocation and alters carbon and nitrogen balance in symbiotic larvae (Table 1). We investigated temperature effects on translocation by examining temperature effects on glucose pools and central carbon metabolism including glycolysis, the tricarboxylic acid cycle, and the pentose phosphate pathway in addition to nitrogen metabolism including assimilation through the glutamine synthetase-glutamate synthase (GS-GOGAT) pathway, the urea cycle, and dipeptide production (Table 1).

**Metabolomics methodological control data analysis.** Our work considered a series of methodological controls. We first examined whether the metabolomic response was impacted by the presence of the $^{13}$C label using the $^{13}$C light versus $^{12}$C light comparisons. To confirm that that the presence of the label did not alter the larval metabolome response to temperature, we conducted a permutational analysis of variance (PERMANOVA) of metabolite pool size from $^{12}$C and $^{13}$C light samples with isotope treatment and temperature treatment as main effects in the *vegan* package [75] using Euclidean distance method with 999 permutations and

**Table 1. Pathways of interest and measured metabolites involved in carbon (glycolysis and central carbon metabolism, pentose phosphate pathway, and tricarboxylic acid cycle) and nitrogen metabolism (ammonium assimilation, urea cycle, and dipeptide production).** References indicate previous reports of metabolomic data that include target pathways in reef-building corals.

| Pathway | Metabolites | References |
|---|---|---|
| Glycolysis and central carbon metabolism | Glucose, glucose-6-phosphate, fructose-6-phosphate, dihydroxyacetone phosphate, 3-phosphoglycerate, pyruvate, lactate | [56,57,67–73] |
| Pentose Phosphate Pathway (PPP) | Ribose-5-phosphate, xylose-5-phosphate | [68,69] |
| Tricarboxylic Acid Cycle (TCA) | Pyruvate, citrate, isocitrate, α-ketoglutarate, succinate, malate | [57,68–70,73] |
| Ammonium assimilation (glutamine synthetase- glutamate synthase; GS-GOGAT) | Glutamine, glutamate, α-ketoglutarate | [56,57,69,72,74] |
| Urea cycle | N-acetylglutamate, carbamoyl phosphate, citrulline, aspartate, arginino-succinate, arginine, ornithine | [70,73,74] |
| Dipeptide production | Arginine-glutamine | [67,74] |

visualized this with principal components analysis (PCA). Further, to verify the presence of the label did not affect pool size of our specific metabolites of interest (Table 1) for carbon (glycolysis and central carbon metabolism, PPP pathway, and the TCA pathway) and nitrogen (GS-GOGAT pathway, urea cycle, and dipeptide production) metabolism, we conducted a two-way ANOVA on pool size with isotope treatment and metabolite as main effects. Post hoc comparisons were conducted between isotope treatment for each metabolite using estimated marginal means via the *emmeans* package in R [76].

Second, to confirm that metabolites of interest were indeed products of photosynthesis that have been translocated to the host or are produced by metabolism of translocated products (and therefore incorporate the $^{13}$C label), we quantified enrichment between $^{13}$C light and $^{13}$C dark samples with the expectation that label incorporation will be minimal or absent in the dark. We performed two-way ANOVA tests for metabolites of interest involved in glycolysis, tricarboxylic acid cycle, and ammonium assimilation (Table 1) with the main effects of metabolite and isotope with post hoc comparisons conducted via the *emmeans* package in R [76].

Finally, to assess the timing of label incorporation in larvae relative to our incubation time, we exposed replicate pools ($n = 10$ larval pools) of *M. capitata* larvae that were reared at ambient temperature to $^{13}$C incubations at the concentrations and light conditions described above across a 24 h time series and sampled at 1, 3, 6, 12, and 24 h. At each time point, larvae were sampled ($n = 2$ pools sampled at each time point) by pouring larvae through sterile 100 μm cell strainers and rinsing 3 times in 1 μm FSW and snap frozen in liquid nitrogen. Samples were stored at −80˚C until processing.

**Metabolomics data analysis.** Following assessment of our methodological controls, we analyzed temperature effects on pool size, enrichment, and carbon-specific enrichment (Fig 2). First, we tested the effect of temperature treatment on metabolomic pool sizes in *M. capitata* larvae using multivariate PERMANOVA tests in the *vegan* package in R [75] to test for differences in metabolite pool size between treatments using unsupervised methods and PCA for visualization. Because these analyses showed significant separation between treatments, we then conducted partial least squares discriminant analysis (PLS-DA) with a two-group model. We followed the PLS-DA with identification of variables of importance in projection (VIP) in the *mixOmics* [77] and *RVAideMemoire* [78] packages in R. VIP scores measure a variable's (metabolite) importance in discriminating between treatments in the PLS-DA model, calculated as the sum of squared correlations between the PLS-DA components and the individual variable. If a metabolite has a VIP score of greater than or equal to 1, it is statistically important in discriminating between pool size variation due to treatment [79]. We then calculated fold change between pool size in each treatment for the identified VIP metabolites and presented

these as mean ± standard error of mean. We then identified the metabolites within pathways of interest (Table 1) within the VIP list. This allowed us to identify carbon and nitrogen metabolism metabolites of interest that had differential pool size between treatment (VIP ≥1; higher pool size in high temperature with fold change >0; lower pool size in high temperature with fold change <0) and those that were not different between treatment (VIP <1).

Second, we followed this same approach to test the effect of temperature treatment on metabolite enrichment between treatments. We first tested the effect of temperature treatment on enrichment using PERMANOVA tests and PCA visualizations followed by PLS-DA analyses and VIP identification as described above. VIPs identified in these analyses indicate metabolites that were statistically important in discriminating between enrichment variation due to treatment groups. We calculated fold change in enrichment between each treatment for VIP metabolites for visualization (mean ± standard error of mean). As done for pool size, we then identified metabolites involved in pathways of interest (Table 1) in the VIP list. This allowed us to identify metabolites of interest that had differential enrichment between treatment (VIP ≥1; higher enrichment in high temperature with fold change >0; lower enrichment in high temperature with fold change <0) and those with no difference in enrichment between treatment (VIP <1).

Third, we analyzed the effect of temperature treatment on the carbon-specific enrichment in metabolites of interest (Fig 2). This was done by conducting two-way ANOVA tests with number of carbons labeled and treatment as the main effects and enrichment as the response variable for each metabolite. *P*-values were adjusted for false discovery rate. Post hoc tests were conducted using estimated marginal means in the *emmeans* package in R [76]. We then plotted the enrichment for each number of carbons for metabolites in our list of metabolites of interest (Table 1).

## Results

### High temperature did not impact larval survival, settlement, physiology, or symbiont photosynthesis, but led to decreased metabolic rates

There were no significant declines in survival (S2A Fig; *P* = 0.677) or settlement (S2B Fig; *P* = 0.670) because of high temperature exposure, but there was 18% mortality across all larvae in the experiment (*P* = 0.012; S1 Table). Larval settlement rates reached a mean of 11% at the end of the study. All statistical results for phenotypic and physiological analyses are presented in S1 Table. Coral larvae exposed to high temperature were significantly larger (7% larger; 0.008 mm$^3$ larger; *P* < 0.001) than those exposed to ambient temperature and this difference was greater than that of our calculated measurement uncertainty (0.00001 mm$^3$).

High temperature larvae exhibited decreased respiration rates (18% decrease on average; *P* = 0.034; Fig 4A). In contrast, there were no differences in net or gross photosynthetic rates (*P* = 0.620 and *P* = 0.133, respectively; Fig 4B and 4C) between temperature treatments. This resulted in a trend for an elevated P:R ratio in exposed larvae due to reduced respiratory rates, but this was not statistically significant (*P* = 0.054; Fig 4D).

Physiological bleaching did not occur in larvae, with no change in symbiont cell densities (*P* = 0.345; S3A Fig), chlorophyll-*a* (protein-normalized *P* = 0.157; cell-normalized *P* = 0.999), chlorophyll-*c2* (protein-normalized *P* = 0.477; cell-normalized *P* = 0.648), or total chlorophyll concentration (protein-normalized *P* = 0.225; cell-normalized *P* = 0.705; S3B and S3C Fig) under high temperature. Although not statistically significant, there was a trend for higher protein-normalized carbohydrate content in exposed larvae (24% greater at high temperature) (*P* = 0.094; S3D Fig). There was no effect of temperature treatment on symbiont cell density-normalized carbohydrate content in the host (*P* = 0.784; S3E Fig).

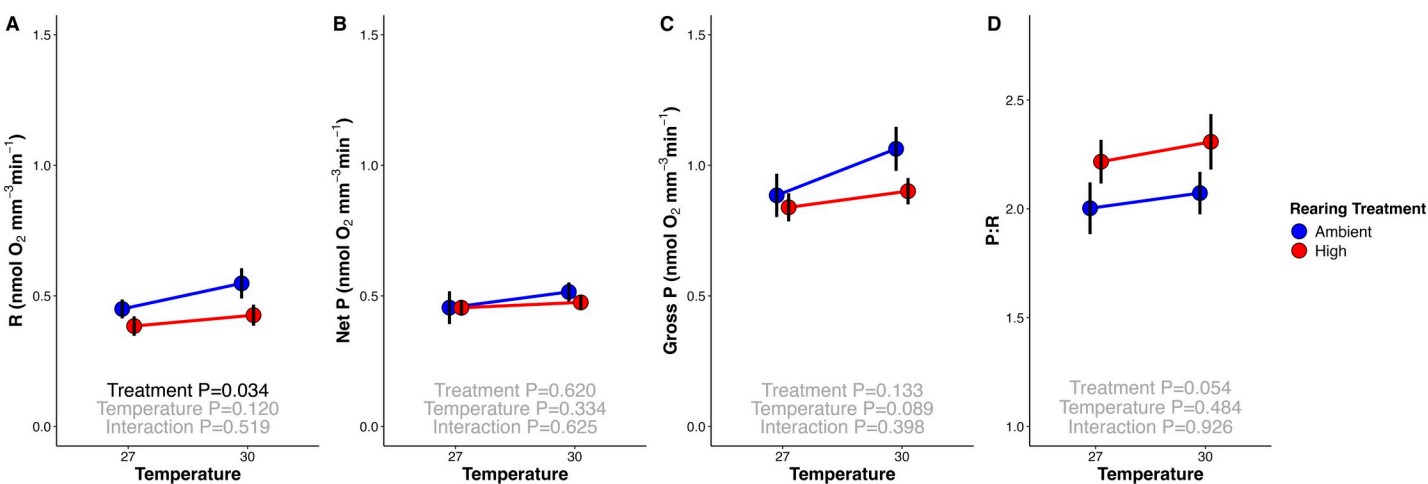

**Fig 4.** *Montipora capitata* **larval mean metabolic rates (nmol oxygen mm$^{-3}$ min$^{-1}$) following 3 days of exposure to ambient (blue) and high (red) thermal rearing treatments measured at control (27˚C) and 30˚C temperatures (x-axis).** (A) Larval respiration (light-enhanced dark respiration; R); (B) net photosynthesis (Net P); (C) gross photosynthesis (Gross P), calculated as the sum of oxygen consumed through respiration and oxygen produced through photosynthesis; and (D) gross photosynthesis:respiration (P:R) ratio. Data normalized to mean larval size from each respective rearing treatment. Error bars represent standard error of mean. Effects of measurement temperature and rearing treatment were tested using two-way ANOVA tests (black text $P < 0.05$; gray text $P > 0.05$). The data underlying this figure can be found at 10.5281/zenodo.13835295.

## Methodological controls of host metabolomic responses

We analyzed differences in metabolomic response between larvae incubated in the light with labeled ($^{13}$C) and unlabeled ($^{12}$C) sodium bicarbonate and those incubated in the dark with the label ($^{13}$C) for metabolites of interest (Table 1). The purpose of these comparisons was to verify: (1) that exposure to the label did not significantly alter response to thermal stress as measured by multivariate metabolite pool sizes (comparison of temperature effects on $^{12}$C and $^{13}$C pool sizes); and (2) that enrichment was higher in the light and minimal to none in the dark for metabolites that we expected to be translocated from the symbiont or generated through metabolism of photosynthetically fixed carbon (comparison of $^{13}$C metabolite enrichment in light versus dark; Table 1).

First, we found that temperature effects were not modulated by isotope treatment, indicated by a nonsignificant interaction between isotope and temperature treatment (PERMANOVA $P = 0.331$; S2 Table). Further, a significant difference in multivariate metabolite pool sizes between ambient and high temperature larvae was detected in both the $^{12}$C (PERMANOVA $P = 0.025$; S4A Fig and S2 Table) and $^{13}$C treatments (PERMANOVA $P = 0.005$; S4B Fig and S2 Table). There was a significant effect of isotope on metabolite pool sizes, regardless of temperature (PERMANOVA $P = 0.001$; S2 Table). Differences in pool sizes of metabolites of interest between $^{12}$C and $^{13}$C exposure included elevated isocitrate (32% higher in $^{13}$C; post hoc $P < 0.001$) and citrate (28% higher in $^{13}$C; post hoc $P = 0.001$) and reduced succinate (15% lower in $^{13}$C; post hoc $P = 0.004$) (S5A Fig). In addition, methionine sulfoxide was elevated with the $^{13}$C label (70% higher in $^{13}$C; post hoc $P < 0.001$). No other carbon (S5A Fig) or any nitrogen metabolism (S5B Fig) metabolites of interest were significantly different between isotope treatments.

Second, we found that glucose and metabolic intermediates of central carbon metabolism contained high $^{13}$C enrichment in the light and minimal enrichment in the dark, demonstrating that $^{13}$C incorporation resulted from photosynthetic fixation of inorganic labeled carbon and translocation to the host. On average, carbon pathway metabolites were enriched

92 ± 0.5% higher in the light than the dark (S6A Fig). All metabolites of interest involved in glycolysis, the pentose phosphate pathway (PPP), and tricarboxylic acid (TCA) cycle were significantly more enriched in the light than the dark (S6A Fig; post hoc $P < 0.001$ for all metabolites). For example, mean (± standard error of mean) enrichment of glucose was approx. 2 orders of magnitude higher at 0.19 ± 0.00 in the light and 0.001 ± 0.00 in the dark (S6A Fig). Metabolites of interest involved in nitrogen metabolism were significantly more enriched in the light than the dark (post hoc $P < 0.01$) with the exceptions of citrulline (post hoc $P = 0.692$), L-arginino-succinate (post hoc $P = 0.029$), arginine (post hoc $P = 0.955$), and ornithine (trend for higher labeling in the light; post hoc $P = 0.084$) (S6B Fig). On average, nitrogen pathway metabolites were 88 ± 1.3% more enriched in the light than in the dark (S6B Fig).

Finally, we collected a set of samples across a 24 h time series to verify that the time of incubation used in this study (4.5 h) allowed for maximum $^{13}$C label incorporation. We observed that peak label incorporation occurred between 3 and 6 h in fructose-6-phosphate, glucose, glucose-6-phosphate, and pyruvate (S7 Fig), supporting our timing of incubation of 4.5 h.

## High temperature impacted multivariate host metabolomic response and metabolite enrichment in *Montipora capitata* larvae

There was a significant effect of temperature treatment on multivariate metabolite pool sizes observed using unsupervised permutational analysis of variance tests (PERMANOVA $P = 0.005$; S4B Fig and S2 Table). Therefore, we used a supervised PLS-DA to identify metabolites that significantly contributed to the metabolomic response to temperature treatment. PLS-DA analyses indicated 28% of the multivariate variance was explained on X-variate 1, describing separation between temperature treatments, with 20% variance explained on X-variate 2, describing variation within treatment (Fig 5A). There were 33 metabolites (VIP ≥1) that significantly contributed to differences in multivariate metabolite pool sizes in response to temperature treatment (S8A Fig and S3 Table). Of the VIP metabolites, 19 were more abundant at high temperature (fold change >0) and 14 were more abundant at ambient temperature (fold change <0) (Fig 5B and S3 Table).

There was a significant effect of temperature treatment on multivariate $^{13}$C enrichment in larvae (PERMANOVA $P = 0.003$; S9 Fig and S2 Table). Supervised PLS-DA analyses indicated 23% of multivariate variance was explained on X-variate 1, describing separation in enrichment between temperature treatments, with 15% variance explained on X-variate 2, describing variation within treatment (Fig 5C). There were 38 VIP metabolites (VIP ≥1) that significantly contributed to differences in multivariate metabolite enrichment in response to temperature treatment (S8B Fig and S4 Table). Of the VIP metabolites, 15 were more enriched at high temperature (fold change >0) and 23 were more enriched at ambient temperature (fold change <0) (Fig 5D and S4 Table). Fold change is indicated as FC hereafter.

## Carbon metabolism: High temperature compromised glycolysis in larvae but glucose translocation remained stable

In this study, we examined treatment effects on carbon metabolism through examination of metabolites involved glycolysis (e.g., glucose), PPP, and TCA pathways listed in Table 1. First, we found a 10% increase in the pool sizes of glucose (VIP = 1.22; FC = 0.10; S3 Table), the metabolic input for glycolysis, in larvae exposed to high temperature compared to those at ambient temperature (Figs 6A and 7A). There was no change in enrichment of glucose (VIP = 0.38; Fig 6B and S5 Table); however, there was a significant difference in carbon-specific enrichment of glucose ($P = 0.004$; Fig 6C and S6 Table). There was a small but significant decrease in the enrichment of glucose molecules that had either 1 (12% lower at high temperature) or 2 (25%

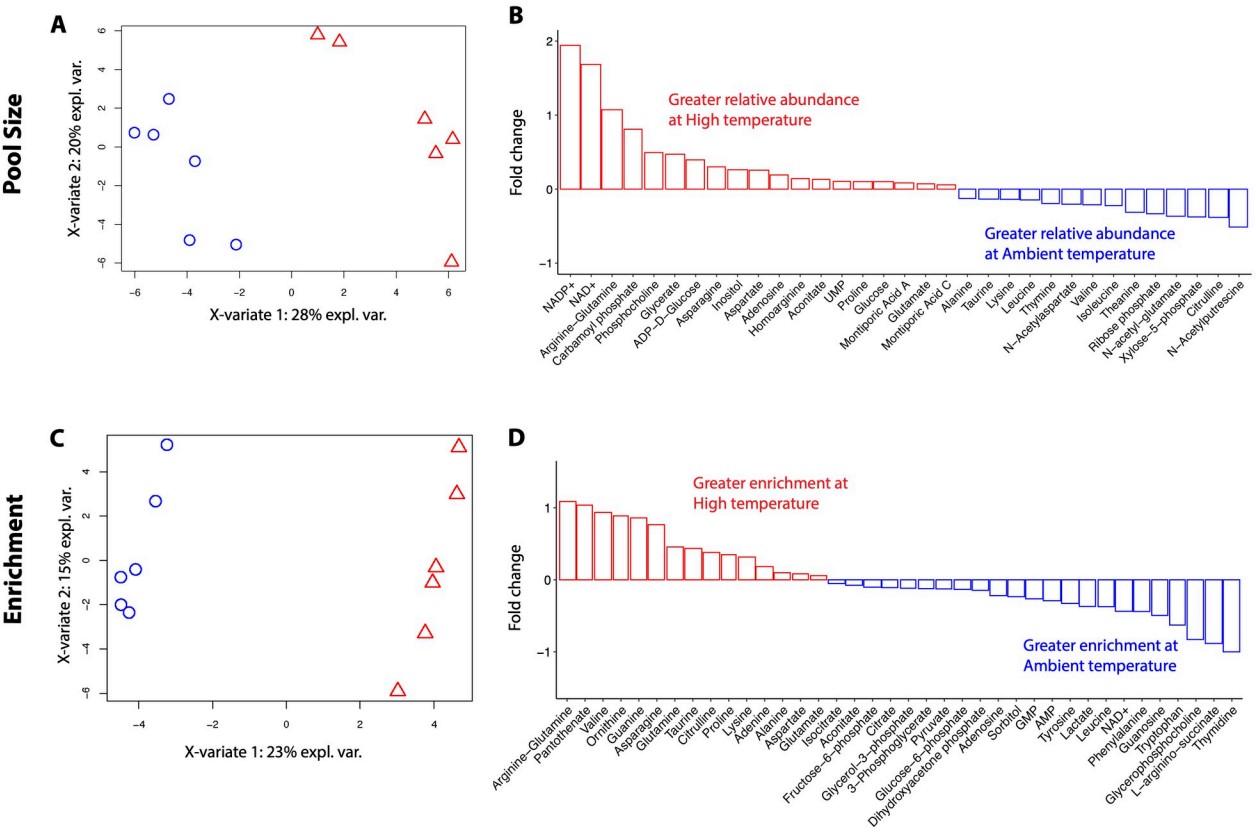

**Fig 5. PLS-DA of metabolite pool size and enrichment.** (A) PLS-DA visualization of metabolite pool size in *Montipora capitata* larvae incubated with [13]C labeled sodium bicarbonate in ambient (blue) and high (red) temperature. (B) Mean fold change in pool size of VIP metabolites driving differences between temperature treatments. Red and blue bars indicate metabolites with larger pools in larvae at high temperature or ambient temperature, respectively. Fold change calculated between mean metabolite value at ambient and mean metabolite value at high temperature. (C) PLS-DA visualization of metabolite enrichment in *Montipora capitata* larvae incubated with [13]C labeled sodium bicarbonate in ambient (blue) and high (red) temperature. (D) Mean fold change in enrichment of VIP metabolites driving differences between temperature treatments. Red and blue bars indicate metabolites with greater enrichment in larvae at high temperature or ambient temperature, respectively. The data underlying this figure can be found at 10.5281/zenodo.13835295. PLS-DA, partial least squares discriminant analysis; VIP, variable importance in projection.

lower at high temperature) [13]C labeled atoms (Fig 6C and S7 Table; post hoc $P_{1C} = 0.001$; post hoc $P_{2C} = 0.019$) with a trend for higher enrichment in glucose molecules with 4 or 5 labeled carbons at high temperature, but this was not significant (Fig 6C; post hoc $P_{4C} = 0.097$).

Through glycolysis, glucose is metabolized into pyruvate through several intermediate steps and metabolites (e.g., glucose-6-phosphate and fructose-6-phosphate; Table 1). The glycolysis pathway metabolites that we measured in this study are visualized in Fig 7A. In analysis of this pathway, we considered metabolites with VIP values ≥1 for pool size and enrichment to be statistically important in distinguishing between treatment groups, with VIP values <1 as not statistically important (S5 and S7 Tables). For pathway visualization, we displayed fold change as percent change (i.e., a fold change of 0.10 is a 10% increase). Except for glucose as described above, glycolysis intermediate metabolites and the product, pyruvate, exhibited no difference in pool size between temperature treatments (VIP <1; S7 Table), but were more enriched in ambient larvae (Fig 7A and S4 Table). Specifically, enrichment was 13% higher in glucose-6-phosphate (VIP = 1.45), 10% higher in fructose-6-phosphate (VIP = 1.13), 15% higher in

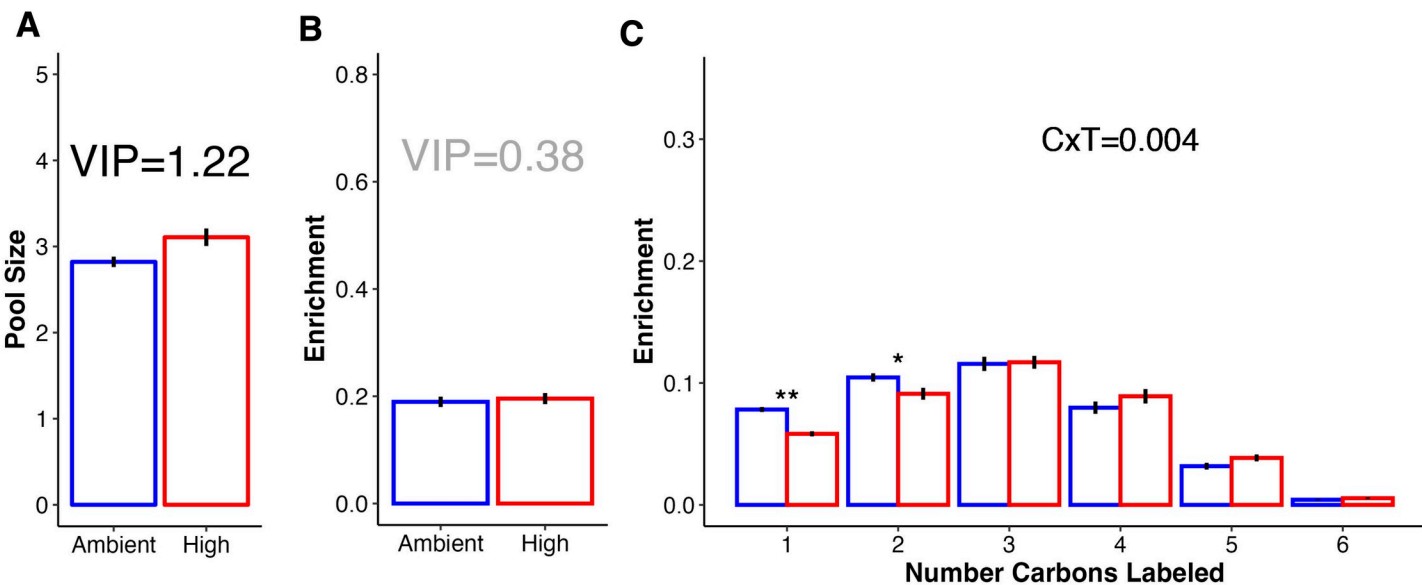

**Fig 6. Glucose pool size and carbon labeling.** (A) Mean (± standard error) pool size (log-transformed median-normalized pool size) and (B) mean (± standard error) $^{13}$C enrichment (proportion) of glucose in larvae at high (red) and ambient (blue) temperature. (C) Mean (± standard error) $^{13}$C enrichment of glucose in *Montipora capitata* larvae shown as enrichment by the number of labeled carbons. Enrichment indicates the probability of the respective number of carbons containing a labeled carbon atom. Significance of the interaction of number of labeled carbons ("C") and temperature treatment ("T") shown in text. In all plots, red indicates high temperature and blue indicates ambient. In (A and B) VIP indicates a variable's importance in projection with VIP ≥1 considered to be statistically important (black text) in contributing to discrimination between treatments with VIP <1 considered to not be statistically important (gray text). In (C), *P*-value indicates significance in two-way ANOVA testing with * indicating estimated marginal means post hoc *P* < 0.05 and ** indicating post hoc *P* < 0.01. No asterisks indicate *P* > 0.05. The data underlying this figure can be found at 10.5281/zenodo.13835295.

dihydroxyacetone phosphate (VIP = 1.43), 12% higher in 3-phosphoglycerate (VIP = 1.06), and 13% higher in pyruvate (VIP = 1.42) at ambient temperature (Fig 7A and S4 Table). There was also higher carbon-specific enrichment in glucose-6-phosphate ($P_{\text{Carbon x Treatment}}$ <0.001; S10A Fig), fructose-6-phosphate ($P_{\text{Carbon x Treatment}}$ <0.001; S10B Fig), and pyruvate ($P_{\text{Carbon x Treatment}}$ = 0.026; S10C Fig and S6 Table).

There was significant accumulation of oxidized energetic cofactors including NAD+ (169% increase, VIP = 1.73) and NADP+ pools (194% increase, VIP = 1.76) in high temperature larvae (Fig 5B and S3 Table) with higher $^{13}$C enrichment of NAD+ in ambient larvae (44% higher, VIP = 1.37; Fig 5D and S4 Table). Relative increases in NADP+ and NAD+ pool sizes were the highest observed for any VIP metabolite (Fig 5B). These cofactors are converted to the reduced form (e.g., NADH, NADPH) as a product of central metabolism. Pyruvate generated in glycolysis can further be metabolized into lactate by lactate dehydrogenase and we found elevated enrichment of lactate at ambient temperature (37% higher, VIP = 1.28; Fig 7A and S4 Table). We also found increased enrichment in glycerol-3-phosphate at ambient temperature (12% higher; VIP = 1.14; S4 Table), which is involved in biosynthesis of triacylglycerol lipids from glycerol and fatty acids. The PPP pathway performs anabolic functions from a glycolysis intermediate, glucose-6-phosphate, generating NADPH and precursors of nucleotide synthesis. Two intermediates of the PPP cycle, ribose phosphate and xylose-5-phosphate were more abundant at ambient temperature (33% and VIP = 1.39; 38% and VIP = 1.67, respectively; S3 Table).

Following glycolysis, pyruvate is metabolized through the TCA cycle to generate cellular energy. We observed higher enrichment in 2 intermediates, citrate (11% higher enrichment,

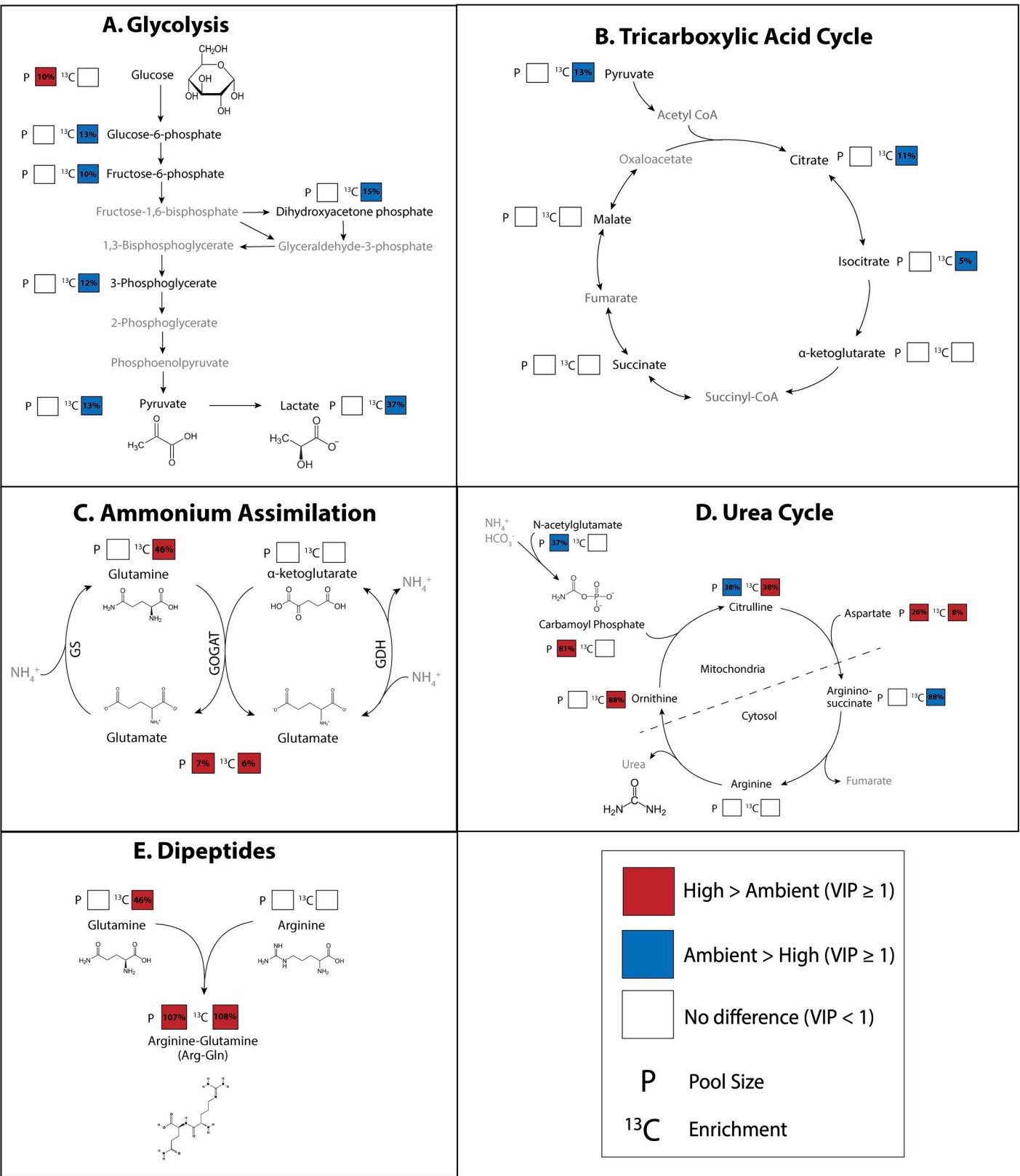

**Fig 7. Metabolite pool size and $^{13}$C enrichment for metabolites involved in key carbon and nitrogen metabolic pathways: (A) glycolysis, (B) tricarboxylic acid cycle, (C) ammonium assimilation (glutamine synthetase (GS), glutamate synthase (GOGAT), and glutamate dehydrogenase (GDH)) pathways, (D) urea cycle, and (E) dipeptide synthesis.** In all plots, black metabolite labels indicate metabolites represented in the data set while gray metabolites were not represented in the data set. Pool size (P) and enrichment ($^{13}$C) squares indicate significant differences between metabolite values in ambient and high temperature exposed larvae (VIP ≥1). White squares indicate no difference between treatments (VIP <1). Red squares indicate the respective value was elevated in high temperature exposed larvae relative to ambient while blue squares indicate value was elevated in ambient larvae relative to those at high temperature. Percentages within squares indicate the percent relative difference in respective metric between treatments.

VIP = 1.13; S4 Table) and isocitrate at ambient temperature (5% higher enrichment, VIP = 1.06; Fig 7B and S4 Table), with no change in pool sizes (VIP = 0.21 and VIP = 0.42, respectively; S7 Table). There was an accumulation of aconitate, an intermediate metabolite during the generation of isocitrate from citrate via aconitase, in the TCA cycle in exposed larvae (13% higher, VIP = 1.04; S3 Table). Further, there was a significant effect of treatment on the number of carbons labeled in aconitate molecules ($P_{\text{Treatment x Carbon}}$ <0.001; S6 Table). There was higher enrichment of aconitate metabolites with only one carbon atom labeled in exposed larvae (post hoc $P$ < 0.001), and more molecules with 3 or 4 labeled carbon atoms in ambient larvae (post hoc $P_{3C}$ = 0.020; post hoc $P_{4C}$ = 0.018; S10D Fig and S6 Table). Differences in metabolite enrichment were not seen further in the progression of the TCA cycle (Fig 7B and S5 Table). It is expected that incorporation of the $^{13}$C label will be highest in the earliest steps of metabolic pathways due to the time required to incorporate the label and detect any significant changes in enrichment in large pools of core metabolic intermediates. Indeed, at ambient temperature, absolute enrichment values early in the pathway including citrate and isocitrate were highest (25.4% and 26.4%, respectively), with values decreasing to 15.3% in succinate and 8.5% in malate.

## Nitrogen metabolism: Increased ammonia assimilation and alterations in host nitrogen sequestration in larvae under high temperature

In this study, we examined treatment effects on nitrogen metabolism through metabolites involved in ammonium assimilation, the urea cycle, and dipeptide production as listed in Table 1. There were significant increases in enrichment of metabolites involved in ammonium assimilation pathways in the host under high temperature. Ammonium assimilation includes the glutamine synthetase (assimilation of ammonium into glutamate to generate glutamine), glutamate synthase/glutamine oxoglutarate aminotransferase (generation of 2 glutamate metabolites from glutamine and α-ketoglutarate), and glutamate dehydrogenase (reversible reaction to assimilate or remove ammonium between glutamate and α-ketoglutarate) pathways (GS, GOGAT, and GDH, respectively) (Fig 7C). We observed increased enrichment in glutamine in high temperature larvae (46% higher, VIP = 2.17; Fig 7C and S4 Table), but no change in pool size (VIP = 0.76; S7 Table). Notably, glutamine had the largest VIP value of all metabolites in driving enrichment differences between treatments (S8B Fig and S4 Table). Glutamate exhibited slightly increased pool size (7% higher, VIP = 1.22; S3 Table) and enrichment (6% higher, VIP = 1.04; S4 Table) in high temperature larvae as compared to ambient larvae (Fig 7C). There was no change in pool size or enrichment of α-ketoglutarate (VIP = 0.34 and VIP = 0.33, respectively; Fig 7C and S5 and S7 Tables).

Temperature treatment also led to changes in the urea cycle in *M. capitata* larvae. In the urea cycle, nitrogen is incorporated into carbamoyl phosphate which then enters the urea cycle and ultimately produces urea, which is excreted by the organism. We observed changes in both pool size and enrichment in metabolites involved in the urea cycle. First, there was significant accumulation of N-acetylglutamate, a precursor to carbamoyl phosphate, at ambient temperature (37% higher, VIP = 1.74) as well as accumulation of citrulline, the first

intermediate of this cycle (38% higher, VIP = 1.68; Fig 7D and S3 Table). In contrast, there was increased enrichment in citrulline (38% higher, VIP = 1.35), aspartate (8% higher, VIP = 1.41), and ornithine (88% higher, VIP = 1.43; Fig 7D and S4 Table) in high temperature larvae. It is possible that some of the total enrichment of citrulline is due to host uptake of inorganic carbon, seen by similar enrichment levels of this metabolite in the dark and the light (S6 Fig). Aspartate exhibited increased pool sizes (26% greater, VIP = 1.53), while citrulline and ornithine exhibited depleted pools (38% decrease, VIP = 1.68; S3 Table) or stable pools, respectively (VIP = 0.04; Fig 7D and S7 Table). Carbamoyl phosphate pools, in contrast, were sharply increased at high temperature (81% higher, VIP = 1.04; S3 Table). There was no change in pool size or enrichment in arginine (VIP = 0.27 and VIP = 0.25, respectively; S5 and S7 Tables) and enrichment of L-arginino-succinate was elevated at ambient temperature (88% higher, VIP = 1.96; Fig 7D and S4 Table).

Finally, there was a large increase in arginine-glutamine (arg-gln) dipeptide pool size and enrichment in coral larvae under high temperature (Fig 7E). There was a disproportionate increase in the pool size (107% increase, VIP = 1.73; S3 Table) and enrichment (108% increase, VIP = 1.78; S4 Table) of arg-gln as compared to pool size and enrichment in arginine and glutamine individually. Specifically, there was no change in enrichment of arginine (VIP = 0.25; S5 Table) and a 46% increase in glutamine enrichment (Fig 7E and S4 Table). Notably, VIP values for arg-gln for both pool size and enrichment were among the highest of all metabolites (S8A and S8B Fig and S3 and S4 Tables).

## Discussion

Our study shows that exposure to increased temperatures compromised central carbon metabolism and disrupted nitrogen homeostasis in *Montipora capitata* larvae (Fig 8). We found that larvae exposed to high temperature displayed decreased glycolysis (Fig 8, step 2), while photosynthetic rates and the translocation of carbon from the symbiont were maintained. Since carbon metabolism decreased and translocation of glucose was stable, there was an accumulation of glucose (10%) in the host (Fig 8, step 1). We also observed that coral larvae invested in nitrogen assimilation and sequestration (Fig 8, step 3), which can facilitate carbon translocation by maintaining the symbiont population in a nitrogen limited state [4,7,22,26,28]. Importantly, coral larvae maintained survival, settlement, and symbiont densities, and did not show physiological signs of bleaching under high temperature. Therefore, our data indicate that larvae employ mechanisms of nitrogen sequestration that can modulate nitrogen availability to the symbiont population to maintain carbon translocation and avoid negative physiological effects of short term (3 day) elevated temperatures.

Larvae in our experiment were exposed to 3 days of high temperature (ambient +2.5°C), reaching mean temperatures of 29.2°C and maximum temperatures of 30.9°C, compared to a mean of 26.7°C and maximum of 27.8°C in the ambient treatment. Corals located on patch reefs within Kāneʻohe Bay, where gametes were collected, experience high diel (1 to 2°C daily temperature fluctuation) and seasonal thermal variability (ranging from annual minimum temperatures of approx. 21°C to annual maximum of approx. 30°C) in the natural environment [48,80]. Further, temperatures during reproductive windows in the summer season in Kāneʻohe Bay (May to August) range from approx. 27 to 30°C, with daily maxima reaching >30°C (2014 to 2023; [48]). Therefore, the high temperature treatment used in this study represents ecologically relevant scenarios experienced during reproductive periods with 1.5 to 2°C warming expected in near-future climate change scenarios [81]. During several recent bleaching events (2014, 2015, and 2019), adult *M. capitata* have shown variability in bleaching response and resilience to bleaching-induced mortality [48,49,82], leading to acclimatization and/or adaptation over the past several decades [83]. Therefore, as coral abundance and cover

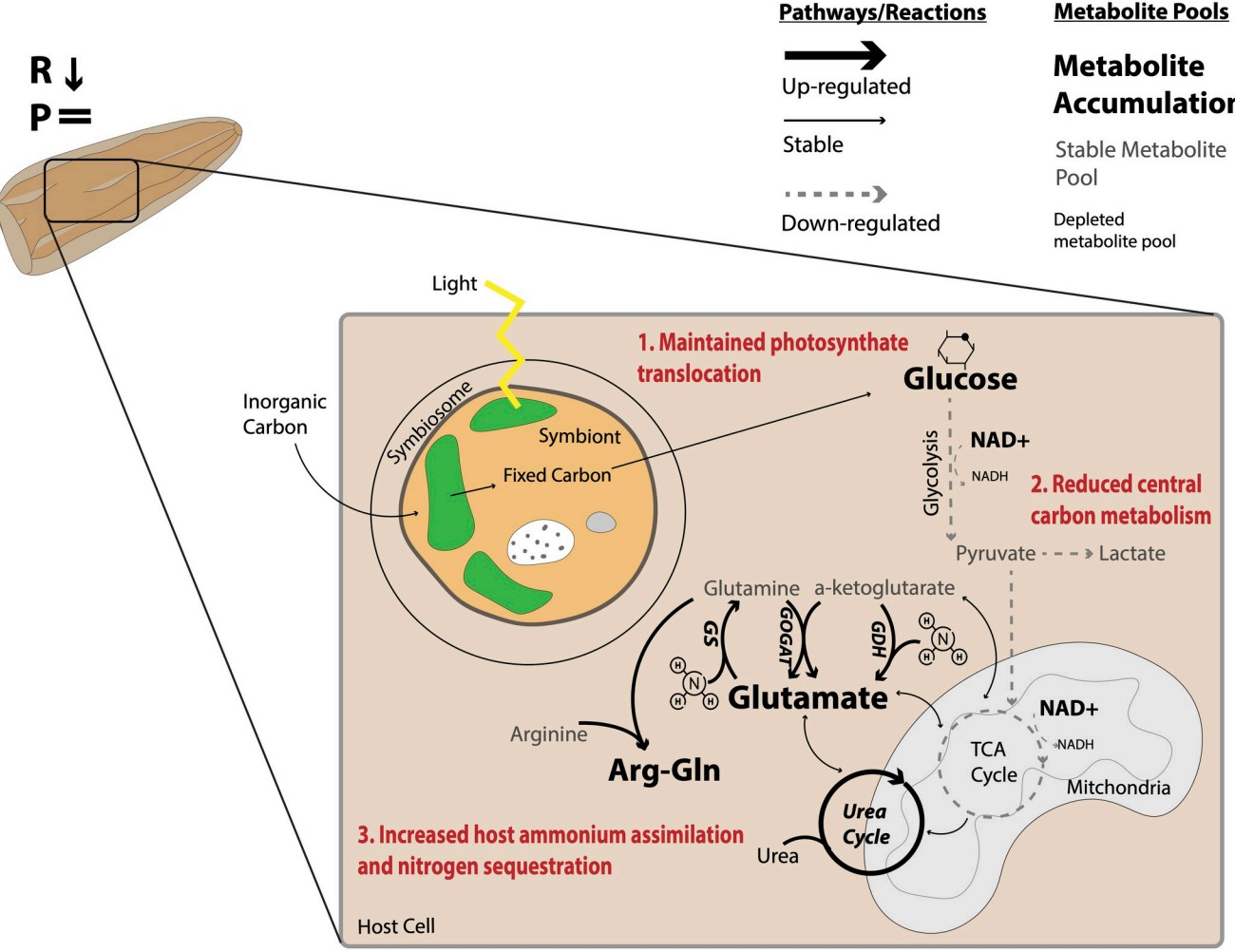

**Fig 8. Conceptual diagram of summarized metabolic response to high temperature in symbiotic larvae.** In this study, we found that exposure to high temperature led to decreased respiration rates (R reduced) in *Montipora capitata* larvae. However, there was no change in photosynthetic rates (P =) and no physiological indications of bleaching or reduced performance (i.e., no change in survival, settlement, symbiont cell densities, or chlorophyll). Although translocation of glucose was maintained (step 1; red text), central carbon metabolism was reduced (step 2; red text), causing glucose accumulation. In response to high temperature, the coral host increased ammonium assimilation and sequestration, which may limit available nitrogen to the symbiont (step 3; red text). Pathways with large, bold arrows were increased while pathways with dotted gray lines were decreased at high temperature relative to ambient. Pathways with solid, thin black lines were stable. Metabolites shown in large, bold letters were accumulated, while those shown in gray were stable and those in small, black text were depleted.

decline and reef functionality transforms under climate change [12,33,84], it is particularly important to study metabolic impacts of temperature in this population of *M. capitata* that inhabits variable thermal regimes and has experienced recent bleaching events.

## Coral larvae decreased metabolic rates and glycolytic flux under high temperature, but carbon translocation was maintained

Under high temperatures, *M. capitata* larvae displayed reduced respiratory rates indicative of metabolic depression, which has also been observed in adult *M. capitata* that experienced

bleaching [85]. Metabolic depression can be a mechanism to conserve energy and resources to facilitate survival and fitness [86,87]. However, the energy reduction due to metabolic depression can incur negative impacts on growth, performance, and reproduction [10,86–89]. Corals are among the organisms that utilize metabolic depression as an acclimatory mechanism while maintaining basal functions required for survival [89]. For example, continued critical functions such as skeletal growth are possible despite a ~2.5-fold reduction in metabolic rate in massive *Porites* spp., because production of skeletal mass is maintained at a maintenance metabolic rate [89]. However, if the stress is prolonged, reserves are consumed and maintenance metabolism is more costly, making metabolic depression unsustainable and additional trade-offs and negative effects can occur [90]. We suggest that reduced metabolic rates in *M. capitata* larvae in this relatively short duration study may have allowed for maintained baseline function while providing energy savings under high temperature. Future work should conduct high frequency time series studies that document the metabolic changes that occur as larvae transition from optimal conditions to moderate, extreme, and lethal levels of thermal stress.

Indeed, we found that performance was maintained in larvae exposed to high temperatures. Our observation that high temperature did not negatively impact larval survivorship corresponds with previous studies that found no difference in *M. capitata* larval survival (4-day exposure to +0.8˚C in [50]) or increases in survival (5-day exposure to +2˚C in [91]) in response to high temperature. Maintained settlement and survival in our study further demonstrate thermal resilience in *M. capitata* larvae to short term, sublethal increases in temperature. However, this resilience may decrease with more severe temperature exposure, for example, as demonstrated by >50% reductions in survival in *M. capitata* larvae exposed to +4˚C for 72 h [92]. Future studies should further investigate the physiological and metabolic trade-offs associated with metabolic depression in coral larvae and evaluate the maximal duration that this strategy can be feasible, including longer term fitness implications (e.g., post-settlement growth, reproduction, and survival).

Reduction in metabolic rate was matched by metabolomic evidence of a decline in central carbon metabolism in high temperature larvae, seen as reduced flux through the glycolysis and TCA pathways. We found higher turnover rates of glycolytic and TCA metabolites at ambient temperature, indicating that central carbon metabolism was reduced in larvae exposed to elevated temperature (Fig 8, step 2). We also found an accumulation of oxidized cofactors (i.e., NAD+ and NADP+) in exposed larvae, further indicating reduced rate of central carbon metabolism because these cofactors are reduced to produce NADH and NADPH through central metabolic reactions. Additional metabolic pathways dependent on glycolysis, including generation of lactate through lactate dehydrogenase activity, were also greater in ambient larvae. Generation of lactate may also help to restore oxidized energetic co-factors that are needed for glycolysis [93]. Declines in central carbon metabolism also occur under stress in adult *Orbicella faveolata* (formerly *Montastraea faveolata)* [94], adult *Mussismilia harttii* [95], and *Pocillopora damicornis* larvae [96]. In contrast, other studies have found increased glycolytic activity under elevated temperature due to temperature-dependent reactions increasing rates of metabolism to meet cellular energy demands in adult *Acropora aspera* [70,97] and *Acropora millepora* [98]. This observed variation in metabolic response to increasing temperature is likely highly dependent on coral and symbiont species, local environmental history, and the duration and magnitude of temperature exposure. We hypothesize that the reduced rates of glycolysis observed in larvae in our study could indicate diversion of cellular resources to other priorities, including host regulation of symbiont population and carbon translocation [7,21,22,27].

Despite reductions in central carbon metabolism, glucose translocation from the symbionts was maintained and symbiont photosynthetic rates were stable across treatments (Fig 8, step

1). This result is consistent with previous work showing maintained glucose translocation under thermal stress and bleaching in symbiotic *Exaiptasia* (previously *Aiptasia sp.*) anemones using [13]C metabolomic tracing [57]. Glucose is a major translocated photosynthate in the coral-algal symbiosis [56,70,99–101]. Glucose enrichment was stable across temperature treatments and was similar to enrichment levels of translocated glucose reported in previous studies (approx. 20% enrichment in both ambient and high temperature) [56,57]. If glucose pools were instead being generated from host catabolism of stored reserves, we would expect the [13]C enrichment of glucose to decrease. This is because carbon stored in lipid reserves is unlabeled [56,100] and, therefore, contributions from stored reserves would reduce [13]C enrichment of the glucose pool, as opposed to increased enrichment that would be expected if glucose is contributed from the symbiont. We observed stable enrichment in glucose in this study, which suggests that the metabolic source of glucose was not altered by high temperature. Future work should further determine photosynthetic tipping points at which temperature impedes or stimulates carbon translocation to the host.

Gluconeogenesis is an alternative metabolic pathway by which glucose is produced from non-carbohydrate pools and can occur during periods of starvation [102] and in corals, during bleaching [103]. However, because gluconeogenesis intermediates (which are the same as those in glycolysis) do not show increased turnover under high temperature and instead are more enriched in ambient conditions, glucose generation through gluconeogenesis is unlikely. Once glucose-6-phosphate is generated from glucose, it can also be diverted to the pentose phosphate pathway, which generates oxidized cofactors and pentose sugars. Two key products of this pathway, ribose-5-phosphate and xylose-5-phosphate, were more abundant at ambient temperature with no change in enrichment and are therefore not likely to be utilized at a higher rate under high temperature. An additional alternative pathway is the glyoxylate cycle or glyoxylate bypass, which generates glucose from lipid storage through beta-oxidation, which has been observed in corals exposed to high temperatures [103–107]. Although glyoxylate was not quantified by our method, metabolites involved in the glyoxylate cycle (which are the same as those in the TCA cycle) including isocitrate, succinate, or malate were equivalent between ambient and high temperature larvae, suggesting that the glyoxylate cycle was not appreciably active. A second line of evidence for this is the expectation that glucose enrichment would decrease if unlabeled lipids were catabolized to produce carbohydrates for further metabolism [56]. Interestingly, one study of the aposymbiotic larval life stage of *Acropora palmata* found an increase in the glyoxylate cycle and peroxisomal lipid oxidation in larvae exposed to high temperature [108]. This suggests that species with aposymbiotic early life stages (i.e., horizontal transmission), rely more heavily on lipid metabolism than vertically transmitting species, but further research and direct comparisons of aposymbiotic and symbiotic life stages is required to address this hypothesis. Given that glucose enrichment was stable across our treatments, we hypothesize that *M. capitata* larvae are not increasing mobilization of lipid stores, nor are they using alternative metabolic pathways, but instead may be using short-term metabolic depression to reduce total energy demand to survive stress.

Larvae at ambient temperature exhibited signs of actively building lipid stores by using glycolysis intermediates and glycerol to generate glycerol-3-phosphate (which was more enriched at ambient temperature) and combining with fatty acids to generate triacylglycerides—a process that is temperature sensitive in corals [98]. This suggests that not only is central carbon metabolism reduced under stress, but larval capacity to build and maintain lipid storage is also compromised. This supports previous observations of reduced [13]C enrichment in host fatty acid pools under high temperature in *Acropora aspera* [56]. However, we did not directly characterize enrichment of lipid or fatty acids pools in this study, and this should be a priority for additional work. Under bleaching scenarios, utilization of lipid stores as an alternative source

of energy is expected when the symbiosis is compromised [105,109,110]. Therefore, a thorough investigation of lipid usage in symbiotic larvae under thermal stress conditions requires carbon tracing experiments targeting lipids and fatty acid pools (e.g., [56]).

The reduction in glycolytic metabolism led to accumulation of glucose in high temperature larvae (Fig 8, step 1). Because symbiont photosynthetic rates were not affected by temperature treatment and the enrichment of glucose was unchanged, increased glucose abundance was not due to increased translocation from the symbiont. This pattern has also been observed in adult *Exaiptasia diaphana* and *Acropora aspera* under high temperature stress in which down-regulation of central carbon metabolism was paired with elevated glucose pools [56,57]. Given these results, we suggest that *M. capitata* larvae down-regulate glycolysis and the TCA cycle, temporarily reducing the demand for cellular energy and diverting energy to maintaining symbiotic homeostasis.

## Symbiotic coral larvae utilize nitrogen assimilation and sequestration to maintain carbon translocation

Nitrogen limitation is critical to maintain a stable coral-Symbiodiniaceae symbiosis [21,22,28]. Larvae in our study elevated ammonium assimilation and nitrogen sequestration, which may act to stabilize the symbiotic relationship (Fig 8, step 3), as evidenced by equivalent symbiont cell densities, chlorophyll, and photosynthetic rates under high temperature. We suggest that coral larvae are therefore able to maintain glucose translocation and avoid negative physiological impacts of thermal stress through nitrogen limitation of the symbiont population. In our study, larvae up-regulated ammonium assimilation through the glutamine synthetase (GS) pathway, indicated by high turnover of glutamine. Since increased ammonium stimulates symbiont metabolic activity and population growth, host assimilation of ammonium into amino acids through the GS-GOGAT (glutamine synthetase-glutamine oxoglutarate aminotransferase; aka glutamine synthetase-glutamate synthase) pathways can be a mechanism of regulating symbiont populations [21,26,30,111]. Both the host and symbiont have the capacity to assimilate ammonium [4,21,28,112,113], although this likely occurs at a greater rate in the symbiont as compared to the host [30]. Nitrogen limitation may favor the translocation of excess photosynthates from the symbiont mediated by changes in symbiont density and population growth [4,7,22,26,28]. However, when nitrogen availability increases and symbiont population growth increases, symbionts may retain more photosynthates for themselves, decreasing translocation to the host [7,9,27,29]. In contrast, previous work in symbiotic *Exaiptasia* anemones found no change in carbon translocation under nutrient limitation induced by starvation [114], demonstrating the need for additional work to clarify how nutrient limitation affects translocation in cases of starvation during feeding life stages, as compared to non-feeding larval stages.

A counter mechanism to changes in GS-GOGAT alone is the usage of glutamate dehydrogenase (GDH), which catalyzes a reversible reaction of ammonium incorporation into α-ketoglutarate to generate glutamate to enable anabolic or catabolic metabolism of amino acids. For example, *Stylophora pistillata* colonies exposed to 21 d of summer maximum heat stress down-regulated GS and GOGAT activity, but increased GDH catabolic activity [28]. Additionally, *Acropora millepora* under acute stress (ambient +4˚C) have been observed to down-regulate GS activity [98] and in *Acropora aspera*, GS was down-regulated at 34˚C [97]. In acute heat stressed *Pocillopora damicornis*, GS and GOGAT activity increased shortly after stress exposure (24 to 26 h at 32˚C), with GDH activity significantly increasing after longer exposure (36 h) [115]. Our observation of increased turnover of glutamine in high temperature larvae suggests that at moderate or sublethal temperatures as used in our study, the coral host favored ammonium assimilation through the GS pathway rather than catabolism through the GDH

pathway. The higher intensity and/or longer duration studies on adults support that under more extreme or prolonged conditions, ammonium metabolism can shift to favor catabolic GDH metabolism to compensate for loss of nutritional resources [28,115]. We suggest that there may be life stage-specific responses to shifts in nitrogen availability in which early life stages favor ammonium assimilation under stress, as compared to adult colonies. Differential use of the GS pathway in larvae compared to adults could be due to relative availability of substrates for catabolism between a single polyp and a colony. Further work is required to address this question by characterizing ammonium assimilation dynamics across coral development in both symbiotic and aposymbiotic life stages.

An emerging mechanism that corals may use for nitrogen balance is the generation of dipeptides as a sink for cellular nitrogen and sequestration to maintain nitrogen limitation of the symbiont (Fig 8, step 3). We observed significantly higher pool size and enrichment of arginine-glutamine dipeptides (arg-gln) in high temperature larvae. The increased production and accumulation of arg-gln has previously been observed as a biomarker of thermal stress in adult corals [67,74]. However, little is known about the synthesis or breakdown of dipeptides in corals and future research should determine whether the primary role is as a sequestration sink, or if these dipeptides perform antioxidant functions, as has been suggested in mice [116,117]. In our study, arg-gln dipeptides were more enriched than either amino acid individually under thermal stress, suggesting active investment in producing arg-gln under elevated temperature [67,74]. Our results corroborate the hypothesis in [74] that production of arg-gln is a possible mechanism of sequestration of nitrogen assimilated into glutamine via GS and we suggest that this is primarily a host-driven response to increased temperature.

*Montipora capitata* larvae in the high temperature treatment also up-regulated the urea cycle, particularly in the mitochondria, which may serve an additional role in sequestering nitrogenous metabolic waste (Fig 7D). Previous work has also found increases in urea cycle activity under stress [74,100] and increased activity in the mitochondria could indicate increased mitochondrial nutrient stress [118]. Under heat stress conditions, *S. pistillata* corals have been shown to increase host assimilation of urea even though assimilation of ammonium is disrupted in both the symbiont and host [119], further demonstrating that relative usage of urea and ammonium metabolism can shift under bleaching conditions. However, because we did not measure concentration of urea ultimately produced through this cycle, additional work is required to determine the relative use of this pathway for nitrogen sequestration through waste products. In addition, excess intracellular nitrogen could be produced by host digestion of symbiont communities, as has been recently demonstrated in adult corals [120]. Digestion of the symbiont community may act to limit symbiont densities and due to increased nitrogen released during digestion would stimulate upregulated assimilation. In early life history when symbiont proliferation is needed to match the energetic demands of the growing host, it is unlikely that symbiotic larvae utilize digestion of symbionts as a mechanism of population control.

The coral holobiont is a complex nutritional network between host, symbiont, and microbial partners and additional work is required to characterize the role of associated microbial communities in affecting carbon-nitrogen balance in corals. Specifically, additional study of the role of diazotroph microbial communities in affecting nitrogen assimilation in the holobiont is required, especially in light of previous work showing that heat stress increases nitrogen fixation by diazotroph communities [121]. Shifts in the relative role of microbial nitrogen fixation that occur as larvae undergo winnowing of microbial partners throughout development [122] may also contribute to life stage specific metabolic responses to heat stress. Applications of stable nitrogen isotope metabolite tracing to coral early life history stages would provide insight into the sources and pathways of nitrogen metabolism across development (e.g., [74]) and the role of microbial communities in contributing to nitrogen metabolism [24].

## Carbon-nitrogen feedback loops in symbiotic coral larvae

Carbon-nitrogen feedback loops are central mechanisms in maintaining a stable symbiosis [21] and explain the larval resilience to physiological bleaching in this study under a +2.5°C 3-day exposure. Carbon that is acquired from the symbiont and metabolized by the host provides the carbon backbones required for nitrogen assimilation into essential amino acids. The interaction between carbon and nitrogen metabolism within the coral-Symbiodiniaceae symbiosis has been described as a feedback loop. Specifically, [21] demonstrated the addition of carbohydrates led to reduced symbiont densities. The authors proposed that increased coral host central carbon metabolism generates metabolites required for ammonium assimilation, leading to increased ammonium assimilation by the host and less available to the symbiont. Reduced ammonium availability for the symbiont limits growth and symbiont photosynthesis [21]. Further, [22] provided evidence that translocation of photosynthates stimulates ammonium assimilation in the host, reducing nitrogen availability and consequently, declines in symbiont population growth rate could favor greater translocation of excess carbon [10,28,29,123].

Our study advances the generality of this carbon-nitrogen feedback loop framework across life stages [21,22,26,113,124], by clearly demonstrating in larvae that symbiont derived carbon is incorporated into amino acids involved in ammonium assimilation. However, there are deviations from the mechanisms described in adult corals. Specifically, we found that the larval coral host increased ammonium assimilation without a concurrent increase in central carbon metabolism, in contrast to previous findings that elevated ammonium assimilation occurs alongside increased carbon metabolism [21,22]. Rather, we observed lower central carbon metabolism paired with higher ammonium assimilation. An alternative explanation of our observation of lower central carbon metabolism paired with higher ammonium assimilation in contrast to previous work could be due to lag effects in metabolic response to symbiont growth and translocation. For example, it is possible that in the early stages of temperature exposure, carbon translocation is reduced, resulting in lower rates of carbon metabolism through glycolysis (Fig 8, step 2). In response to decreased translocation, energy savings could then be diverted to ammonium assimilation and dipeptide synthesis (Fig 8, step 3), leading to reduced symbiont population growth and stimulation of carbon translocation back to normal levels (Fig 8, step 1). Following return to homeostasis, the host would then be able to restore normal carbon metabolism rates, but it is possible that we did not yet observe this change in the temporal scale of our sampling. Determination of the temporal unfolding of this mechanism would require a high frequency time series from the initial to the bleaching states. Further, the contrast between our results and previous work may be due to differences in metabolic responses to stress in larvae as compared to adult corals and requires additional work exploring the mechanisms of carbon-nitrogen feedback loops across life stages.

Overall, the results of our study demonstrate that metabolic feedback and nitrogen limitation play important roles in surviving thermal stress in symbiotic larvae. Not only do coral larvae receive nutrition from their symbionts, but symbiotic nutrition is an important component of metabolic activity. As sea surface temperatures continue to increase, it is vital to understand how early life history symbiotic relationships are affected and determine metabolic tipping points impacting successful coral development.

## Supporting information

**S1 Fig. Temperature treatments (ambient = blue; high = red) during each period of the study: embryonic development (dotted line; 06–12 through 06–14), larval exposure (dashed line; 06–15 through 06–18), and settlement (solid line; 06–19 through 06–26).** X-

axis indicates date (MM-DD). Asterisks (*) indicate the start of temperature treatments (06–14) and plus (+) indicates time of larval sampling (06–18). Temperature recorded every 15 min by $n = 3$ loggers per temperature treatment. Each point represents individual temperature measurements recorded by each logger. The data underlying this figure can be found at 10.5281/zenodo.13835295.
(TIFF)

**S2 Fig.** *Montipora capitata* **larval (A) survival during 3 days of exposure to ambient (blue) and high (red) temperature treatments.** (B) Percent larval settlement. Following larval exposure, larvae were settled under temperature treatments and held at these treatments over a 5-day period. Effects of time and rearing treatment were tested using linear mixed effect models with tank as a random intercept (black text $P < 0.05$; gray text $P > 0.05$). In both plots, points represent individual replicates. Linear model fit lines shown with gray indicating 95% confidence intervals. The data underlying this figure can be found at 10.5281/zenodo.13835295.
(TIFF)

**S3 Fig. Box plots of** *Montipora capitata* **larval physiological measurements following 3 days of exposure to ambient (blue) and high (red) temperature treatments.** (A) Symbiont cell density, cells per μg host protein content. (B) Symbiont chlorophyll ($a + c2$), μg per μg host protein content. (C) Symbiont chlorophyll ($a + c2$), μg per symbiont cell. (D) Host carbohydrate content, μg per μg host protein content. (E) Host carbohydrate content, μg per symbiont cell. For all responses, effect of treatment was tested using Welch $t$ tests (black text $P < 0.05$; gray text $P > 0.05$). In all plots, point represent individual replicates. The data underlying this figure can be found at 10.5281/zenodo.13835295.
(TIFF)

**S4 Fig. Multivariate visualization of metabolite pool size in** *Montipora capitata* **larval samples.** (A) Principal components analysis of metabolite pool sizes in larvae incubated with unlabeled ($^{12}$C) sodium bicarbonate between temperature treatments. (B) Principal components analysis of metabolite pool sizes in larvae incubated with labeled ($^{13}$C) sodium bicarbonate between temperature treatments. Blue indicates larvae exposed to ambient temperature; red indicates larvae exposed to high temperature. Axis show percent variance explained by each principal component. *P*-values indicate significance of temperature treatment on multivariate pool size analyzed using PERMANOVA analyses. The data underlying this figure can be found at 10.5281/zenodo.13835295.
(TIFF)

**S5 Fig. Mean (± standard error of mean) pool size in metabolites of interest related to (A) carbon metabolism, including glycolysis, pentose phosphate pathway, and the tricarboxylic acid cycle.** (B) Mean (± standard error of mean) pool size in metabolites of interest related to nitrogen metabolism, including ammonium assimilation and the urea cycle in $^{12}$C (gray) and $^{13}$C (green) isotope treatments. In all plots, error bars represent standard error of mean. ** indicates $P < 0.01$ as determined by estimated marginal means post hoc tests. No asterisks indicate $P > 0.05$. The data underlying this figure can be found at 10.5281/zenodo.13835295.
(TIFF)

**S6 Fig. Enrichment in metabolites of interest in** $^{13}$**C light and dark treatments.** Mean (± standard error of mean) enrichment in metabolites of interest related to (A) carbon (including glycolysis, pentose phosphate pathway, and the tricarboxylic acid cycle) and (B) nitrogen metabolism (including ammonium assimilation and the urea cycle) in larvae incubated with

labeled $^{13}$C sodium bicarbonate in the light (green) and the dark (blue). * Indicates $P < 0.05$ and ** indicates $P < 0.01$ as determined by estimated marginal means post hoc tests. No asterisks indicate $P > 0.05$. In (A), all metabolites were significant at $P < 0.001$. The data underlying this figure can be found at 10.5281/zenodo.13835295.
(TIFF)

**S7 Fig. *Montipora capitata* larval stable isotope ($^{13}$C sodium bicarbonate; 4 mM) metabolomic time series methodological control.** Isotopic label enrichment shown for glucose (a primary photosynthate), glycolysis intermediates (fructose-6-phosphate, glucose-6-phosphate), and a glycolysis product (pyruvate) across a 24 h time series sampling. Points represent individual replications. The data underlying this figure can be found at 10.5281/zenodo.13835295.
(TIFF)

**S8 Fig. Metabolite Variable Importance in Projection (VIP) scores for metabolites with VIP scores ≥1 that significantly drive differences in (A) pool size and (B) $^{13}$C enrichment between high and ambient temperature metabolomic responses in *Montipora capitata* larvae.** VIPs were identified through partial least squares discriminant analysis (PLS-DA). The data underlying this figure can be found at 10.5281/zenodo.13835295.
(TIFF)

**S9 Fig. Multivariate visualization principal components analysis of metabolite $^{13}$C enrichment in *Montipora capitata* larvae exposed to ambient (blue) and high (red) temperatures.** Axes show percent variance explained by each principal component. *P*-value indicates significance of temperature treatment on multivariate enrichment analyzed using PERMANOVA analyses. The data underlying this figure can be found at 10.5281/zenodo.13835295.
(TIFF)

**S10 Fig. Mean (± standard error of mean) metabolite enrichment for molecules of each metabolite with the specified number of labeled carbons.** Significance (*P*-value) of the interaction of number of labeled carbons ("C") and temperature treatment ("T") shown in text determined by two-way analysis of variance tests. In all plots, red indicates high temperature and blue indicates ambient temperature. Asterisks indicate significance of post hoc comparisons with * indicating $P < 0.05$ and ** indicating $P < 0.01$. No asterisks indicate $P > 0.05$. The data underlying this figure can be found at 10.5281/zenodo.13835295.
(TIFF)

**S1 Table. Statistical analyses of phenotypes, metabolic rates, symbiont physiology, and carbohydrate content in *Montipora capitata* larvae reared at ambient and high temperatures.** Survival and settlement were analyzed using linear mixed effect models with tank as a random intercept. Metabolic rates analyzed using or two-way analysis of variance tests. Physiological metrics analyzed using two-sample Welch *t* tests except for holobiont carbohydrates per symbiont cell analyzed with a Wilcoxon rank sum exact test due to violation of homogeneity of variance. Bold indicates $P < 0.05$.
(XLSX)

**S2 Table. Permutational multivariate analysis of variance (PERMANOVA) of metabolite pool size for all samples with isotope, treatment, and their interaction as main effects.** PERMANOVA analysis of metabolite pool size ($^{12}$C unlabeled and $^{13}$C labeled samples) and metabolite enrichment ($^{13}$C labeled samples) in *Montipora capitata* larvae reared at ambient and high temperatures with treatment as the main effect. Bold indicates $P > 0.05$. DF indicates degrees of freedom and SS indicates sum of squares.
(XLSX)

**S3 Table. Mean and standard error of mean (SE) pool size (log-transformed median-normalized pool size) of variable importance in projection (VIP) metabolites in ambient and high temperature treatments.** Relative difference calculated between ambient and high temperature treatments as fold change. Positive values indicate larger pool size at high temperature (bolded text), with negative values indicating larger pool size at ambient temperature. All metabolites shown here are considered statistically important in discriminating between treatments at VIP ≥1.
(XLSX)

**S4 Table. Mean and standard error of mean (SE) enrichment (proportion) of variable importance in projection (VIP) metabolites in ambient and high temperature treatments.** Relative difference calculated between ambient and high temperature treatments as fold change. Positive values indicate greater enrichment at high temperature (bolded text), with negative values indicating greater enrichment at ambient temperature. All metabolites shown here are considered statistically important in discriminating between treatments at VIP ≥1.
(XLSX)

**S5 Table. Variable importance in projection (VIP) for enrichment of metabolites with VIP values of <1.** Metabolites with a VIP <1 are considered not statistically important in discriminating between treatment groups.
(XLSX)

**S6 Table. Analysis of variance (ANOVA) tests for the effect of treatment and number of carbons labeled for each metabolite in labeled carbon positions.** DF = degrees of freedom; SS = sum of squares; FDR = false discovery rate adjusted *P*-value. Bold indicates adjusted $P < 0.05$.
(XLSX)

**S7 Table. Variable importance in projection (VIP) of pool size (log-transformed median-normalized pool size) for metabolites with VIP values of <1.** Metabolites with a VIP <1 are considered not statistically important in discriminating between treatment groups.
(XLSX)

## Acknowledgments

As guests, we recognize and give thanks for the land and water resources of the ʻāina and the traditional owners of the land, kānaka ʻōiwi, both past and present, as well as future generations, on which this experimental work was conducted in the Kāneʻohe Ahupuaʻa and the islands of Hawaiʻi. We are grateful for the logistical support provided by the Coral Resilience Lab and the Hawaiʻi Institute of Marine Biology. We gratefully acknowledge logistical and experimental support from Amanda Williams, Luella Allen-Waller, Chris Suchoki, and the Coral Ecology Lab and Coral Resilience Lab at the Hawaiʻi Institute of Marine Biology. Thank you to Kevin Wong for guidance on metabolomics extractions and to the Putnam Lab (University of Rhode Island) and Roberts Lab (University of Washington) for feedback on earlier versions of this manuscript.

## Author Contributions

**Conceptualization:** Ariana S. Huffmyer, Hollie M. Putnam.

**Data curation:** Ariana S. Huffmyer, Jill Ashey, Emma Strand, Eric N. Chiles, Xiaoyang Su, Hollie M. Putnam.

**Formal analysis:** Ariana S. Huffmyer, Jill Ashey, Emma Strand, Eric N. Chiles, Xiaoyang Su.

**Funding acquisition:** Ariana S. Huffmyer, Hollie M. Putnam.

**Investigation:** Ariana S. Huffmyer.

**Methodology:** Ariana S. Huffmyer, Jill Ashey, Eric N. Chiles, Xiaoyang Su.

**Project administration:** Ariana S. Huffmyer.

**Resources:** Hollie M. Putnam.

**Supervision:** Hollie M. Putnam.

**Validation:** Ariana S. Huffmyer.

**Visualization:** Ariana S. Huffmyer, Hollie M. Putnam.

**Writing – original draft:** Ariana S. Huffmyer.

**Writing – review & editing:** Ariana S. Huffmyer, Jill Ashey, Emma Strand, Eric N. Chiles, Xiaoyang Su, Hollie M. Putnam.

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
