## [Editor Report · Decision Letter 0]

19 Jun 2024

Dear Dr Huffmyer, 

Thank you for submitting your manuscript entitled "Coral larvae employ nitrogen sequestration mechanisms to stabilize carbon provisioning from algal symbionts under increased temperature" for consideration as a Research Article by PLOS Biology.

Your manuscript has now been evaluated by the PLOS Biology editorial staff, as well as by an academic editor with relevant expertise, and I'm writing to let you know that we would like to send your submission out for external peer review.

Once your full submission is complete, your paper will undergo a series of checks in preparation for peer review. After your manuscript has passed the checks it will be sent out for review. To provide the metadata for your submission, please Login to Editorial Manager (https://www.editorialmanager.com/pbiology) within two working days, i.e. by Jun 21 2024 11:59PM.

Kind regards,

Roli Roberts

Roland Roberts, PhD

Senior Editor

PLOS Biology

rroberts@plos.org

---

## [Decision Letter · Decision Letter 1]

8 Sep 2024

Dear Dr Huffmyer,

Thank you for your patience while your manuscript "Coral larvae employ nitrogen sequestration mechanisms to stabilize carbon provisioning from algal symbionts under increased temperature" was peer-reviewed at PLOS Biology. It has now been evaluated by the PLOS Biology editors, an Academic Editor with relevant expertise, and by several independent reviewers. 

Based on the reviews and on our Academic Editor's assessment of your revision, we are likely to accept this manuscript for publication, provided you satisfactorily address the remaining points raised by the reviewers regarding the text and figure modifications. Please also make sure to address the following data and other policy-related requests, as well as the following editorial requests:

a) We routinely suggest changes to titles to ensure maximum accessibility for a broad, non-specialist readership, and to ensure they reflect the contents of the paper. In this case, we would suggest a minor edit to the title, as follows. Please ensure you change both the manuscript file and the online submission system, as they need to match for final acceptance:

"Coral larvae increase nitrogen assimilation to stabilize algal symbiosis and combat bleaching under increased temperature"

Please supply the numerical values either in the a supplementary file or as a permanent DOI’d deposition for the following figures:

Figure 3B, 4ABCD, 5ABCD, 6, S1, S2AB, S3ABCDE, S4AB, S5AB, S6AB, S7, S8, S9, S10

c) Please cite the location of the data clearly in all relevant main and supplementary Figure legends, e.g. “The data underlying this Figure can be found in S1 Data” or “The data underlying this Figure can be found in https://doi.org/10.5281/zenodo.XXXXX”

d) Please ensure that your Data Statement in the submission system accurately describes where your data can be found and is in final format, as it will be published as written there.

e) Many thanks for providing the underlying code in GitHub. However, because Github depositions can be readily changed or deleted, please make a permanent DOI’d copy (e.g. in Zenodo or OSF) and provide this URL in the manuscript and Data Availability Statement.

We expect to receive your revised manuscript within two weeks. 

*Published Peer Review History*

*Press*

Sincerely,

Melissa 

Melissa Vázquez Hernández, PhD

Associate Editor, PLOS Biology

on behalf of

Roland

Roland Roberts, PhD

Senior Editor

rroberts@plos.org

PLOS Biology

REVIEWERS' COMMENTS:

Reviewer #1: 

The manuscript assesses the role of nutrient cycling dynamics in coral larvae under heat stress. Also, this is a topic that is well studied in adult corals, we are missing baseline data in coral larvae. Search, this manuscript fills an important knowledge gap. The study is very well embedded into the current literature of carbon/nitrogen cycling between coral host and algal symbiont, and how the 'balanced tranfser' is maintaining a stable symbiosis. in this regard, the findings are remarkable: that larvae 'manipulate' the nitrogen availability to keep algal symbionts 'in check'.

What might be useful to add is a discussion on the role of diazotrophs. Admittedly, this study is already vast, so I would not see the necessity to do 16S/diazotroph profiling. However, based on the current literature (e.g., https://academic.oup.com/ismej/article/16/4/1110/7474332), change of diazotroph community/activity should be mentioned in the Discussion. This might be particularly interesting in light of a 'maturing' microbiome (winnowing = larvae respond differently than adult counterparts), and the putative high level of species specificity in the heat stress response, as nicely pointed out in the discussion

I was surprised to see that the GitHub repo is not yet publicly available (what harm could be done?), so I trust the authors will make sue everything is not only uploaded to archives but also publicly accessible. 

It's a rather lengthy read that, however, exercises excellent integration with the available literature (aside from the diazotroph aspect, see above).

Reviewer #2: 

This manuscript describes a detailed metabolomics study of coral larvae under thermal stress. The findings add nicely to those form previous metabolomics studies of corals and other symbiotic cnidarians, both under thermal stress and in response to different Symbiodiniaceae species, by improving our understanding of how the early life stages of corals might respond to, and combat, elevated seawater temperatures. Overall, the manuscript is well written, with an excellent level of detail. The manuscript is quite long (the methods especially so) and, as tends to happen with such large datasets, it takes the reader a little while to grasp all of the complex interactions being described - however, I don't think that this requires any changes from the authors. Overall, I have relatively few comments on the science, as the authors have done an excellent job of explaining and discussing the findings, and I support the manuscript's publication pending a few minor edits:

Line 59: Better as "Symbiodiniaceae, which ARE"

Line 70: Should be "of lipid reserves..."

Lines 89-91: The grammar here is a little odd - what is meant by "time window" (is this even necessary?), while "in" on Line 91 is an error I think.

Line 105: Should be "increases...impact metabolism"

Line 115: "Falcon" should start with a capital letter.

Line 138: Delete "was" to say "treatment that supplied..."

Lines 145-149: This part is unclear - I suggest trying to clarify what's meant by the ambient and high temperature treatments - for instance, why are two mean temperatures and sample sizes stated for the ambient treatment in Lines 148-149 - is some information missing here? The same query arises in Lines 241-242.

Line 149 (and elsewhere): "PAR" isn't a unit - replace this with "umol photons m-2 s-1". 

Line 168: Should be "sensor dish readerS"

Line 179: End bracket is superscripted

Line 192: Delete "were"

Line 217: Do you mean "conical flasks"?

Line 220: Should be "IS sourced..."

Line 250 (and elsewhere in the methods, e.g. Lines 265-266): Should be 600 ul...WERE added..."

Line 253: Better as "designated AS the..."

Line 254: Better as "for MEASURING symbiont cell..."

Lines 299, 303 and 499: "c2" should be italicized.

Line 495: How is it possible that Pgross didn't change, R did change, and therefore that there was no change in Pnet given that this parameter is a function of the other two? Was this simply a statistical quirk or is there an error?

Line 499: Chlorophyll-a should use an italicized "a", not alpha.

Lines 727-728: The grammar is a bit wayward here, so needs to be tidied up.

Lines 738-739: A similar conclusion about maintained glucose/C translocation during bleaching was reached by Hillyer et al. (2017) New Phytologist. This paper is cited elsewhere, but it's worth noting this particular point here.

Line 753: Delete "therefore"

Line 767: I don't think it quite right to refer to this as an "aposymbiotic species" or "aposymbiotic coral" - what you mean is a coral with an aposymbiotic stage in its life-history. Perhaps clarify what you mean here.

Line 787: "Aiptasia sp" is old taxonomy - replace this with "Exaiptasia diaphana".

Line 789: Better as "M. capitata corals downregulate glycolysis".

Lines 807-810: There has been a tendency to state this mechanism (N limitation for promoting C translocation) as a fact in the last few years, yet the only study (though I could be wrong!) to directly test this hypothesis (Davy & Cook 2001, Marine Biology) reported no change in %translocation with N limitation. While much more study is still needed on this topic, I suggest some less definitive wording here, and citation of the aforementioned paper, which is currently absent.

Discussion section B: A recent study that may or may not be relevant to this part of the discussion is Crehan et al. 2024, J. Exp. Biol., that describes the impacts of thermal stress on the assimilation of inorganic and organic N sources.

Line 837: Should it say "THOSE of" rather than "that of"?

Line 862: Better as "THE larval" rather than "our larval".

Overall, this is a very nice manuscript!

Reviewer #3: 

The authors attempt to determine how heat impacts coral host metabolism during heat stress. Overall, the manuscript is a delight to read and has important data for the field. The writing is done well and handles a complex metabolic experiment with ease. Specifically, the authors examine metabolite levels and 13C incorporation in symbiotic M. capitata larvae after 3.5 days of either ambient or high temperature. They chose temperatures that are 'ecologically relevant'; however, I will note that this is not needed for good laboratory experiments. The high-temperature treatment had no impact on survival, symbiont density, or algal photosynthesis, suggesting that the animals are not bleaching during this window. The authors find that the heat-exposed larvae have an overall decrease in metabolism. This conclusion is supported by lowered respiration of the host and decreased 13C enrichment in glycolysis and TCA cycle intermediates. Even though metabolism is disrupted, carbon translocation from the algae to the host is sustained. Additionally, this translocated carbon is found enriched in glutamine and glutamate because of increased ammonium assimilation. Overall, the authors conclude that the host reserves/increases its glucose levels under these heat conditions and maintains its nutritional symbiosis with the algae by increasing ammonium assimilation and nitrogen sequestration.

Major comments: 

The manuscript is overall good. However, I recommend some minor changes that may be helpful to the reader. 

First, the paper's title suggests there is data showing that nitrogen sequestration mechanisms are essential to stabilizing carbon provisioning from algal symbionts under increased temperature. I believe this was more of a hypothesis generated in the discussion. The title, as is, oversteps the data.

Second, in the abstract, Lines 37-39 also overstep the data. There are no functional data supporting this claim. This conclusion appears to be a hypothesis from the data. Also, in Lines 40-42, this conclusion is based on the correlation between the metabolic trends and the lack of bleaching. These changes may or may not impact bleaching. Minor changes to the abstract and discussion would help highlight some degree of uncertainty. 

Third, a key finding in the paper, Discussion Lines 873-888, seems to conflict with refs 21 and 22. The difference could be due to different coral life stages, species, or other reasons. As these results challenge major correlative conclusions made by the other references, these conflicting results should be a key point of the paper and the abstract. 

Minor comments:

In the introduction, the breakdown of nutrient exchange is treated as a known mechanism driving bleaching. However, this has not been formally tested and remains a model. There are other models as well. It is worth clarifying that it is a model (e.g., lines 58-63). 

Similarly, conclusions in Lines 64-75 could be made stronger by highlighting which are strongly supported by the data vs. hypothesized mechanisms. The data in the paper fill in some of these holes. However, it is unclear where the outstanding questions are from reading the introduction. 

In Figure 1, there is no indication of the isotope treatment along the timeline of the experiment. 

Line 496: What do the authors mean by "trend for an elevated P:R ratio".

Line 504: Has a test for normality been done on S3D? It looks like the data might be bimodal, suggesting a different statistical test be used. 

Line 517: No reference to the data.

Line 450: Should there be a citation for the multivariate PERMANOVA tool?

Line 519 States a significant difference in the multivariate metabolite pool sizes between ambient and high-temperature larvae for both 12C and 13C. However, when referencing the S5A figure, there is no comparison of temperature. I believe they meant to refer to S4A. 

Line 522: No reference to supplemental data.

Line 605: no reference to the data.

Line 607-609: inconsistent referencing to data. 

Line 670: citation for maintaining the symbiont population in a nitrogen-limited state.

Line 842: "… in the high [] treatment.."

Line 842: no reference to figure for the upregulation of the urea cycle, particularly the mitochondria. 

Line 871: "…growth rate declines this would allow for…" needs to be rewritten.

---

## [Editor Report · Decision Letter 2]

1 Oct 2024

Dear Ariana,

Thank you for the submission of your revised Research Article "Coral larvae increase nitrogen assimilation to stabilize algal symbiosis and combat bleaching under increased temperature" for publication in PLOS Biology. On behalf of my colleagues and the Academic Editor, Nancy Moran, I'm pleased to say that we can in principle accept your manuscript for publication, provided you address any remaining formatting and reporting issues. These will be detailed in an email you should receive within 2-3 business days from our colleagues in the journal operations team; no action is required from you until then. Please note that we will not be able to formally accept your manuscript and schedule it for publication until you have completed any requested changes.

IMPORTANT: I've asked my colleagues to include the following request among their own: "Many thanks for including the numerical values underlying your Figures in your Zenodo deposition. However, in the Fig legends you currently cite the OSF DOI. Please could you change your legend citations to the Zenodo URL/DOI? (or copy the Fig-specific data files across to OSF, but that seems more troublesome)."

Sincerely, 

Roli

Senior Editor

PLOS Biology

rroberts@plos.org